# HOW TO DISTILL TASK-AGNOSTIC REPRESENTATIONS FROM MANY TEACHERS?

## ABSTRACT

Casting complex inputs onto tractable representations is a critical step in many fields. Differences in architectures, loss functions, input modalities, and datasets lead to embedding models that capture diverse information of the input. Multi-teacher distillation seeks to exploit this diversity to create richer representations but often remains task-specific. We extend this framework by proposing a task-oriented setting that introduces an objective function based on the "majority vote" principle. We demonstrate that the mutual information between the student and the teachers is an upper bound for this function, providing a task-agnostic loss for our distillation procedure. An extensive evaluation is performed in different domains —natural language processing, computer vision, and molecular modeling — indicating that our method effectively leverages teacher diversity to produce more informative representations. Finally, we use our method to train and release new state-of-the-art embedders, enabling improved downstream performance in NLP and molecular modeling.

## 1 INTRODUCTION

Casting complex inputs into tractable representations is essential for many applications in different fields, from natural language processing (Li & Li, 2023; Pimentel et al., 2023), computer vision (Kubota et al., 2024; Bhalla et al., 2024; Khandelwal et al., 2022) to bioinformatics (Morgan, 1965; Rogers & Hahn, 2010; Wang et al., 2022a). This is done using embedders that project an object (image, text, molecules,...) into numerical representations, enabling various downstream tasks (Murphy, 2013; Vilnis & McCallum, 2015).

There are a variety of architectures, training settings (unsupervised, supervised, etc.), objective functions (masked language modeling, contrastive learning, etc.), and datasets used for embedders. Large pretrained models have recently become a natural starting point to create embedders (Che et al., 2024; Touvron et al., 2023; Jiang et al., 2023; Meng et al., 2024). Every combination of methods has its strengths and weaknesses, leading to embedders that capture slightly different information about the input.

To leverage the diversity of these representations, a common practice is to combine them into a single model, a process commonly known as Multi-Teacher Knowledge Distillation (Hinton et al., 2015; Zhang et al., 2023). Not only are these methods cost-effective (Hinton et al., 2015; Frosst & Hinton, 2017), they are also extremely useful to pack more information into smaller models from bigger ones (Pan et al., 2022; Wang et al., 2023; Zhang et al., 2023), or mend the weights of models whose architectures have been altered (Muralidharan et al., 2024). However, most of these works focus on distilling representations to solve a single task, whereas we are interested in building general representations.

To the best of our knowledge, there are few methods that address task-agnostic representation distillation in the context of multi-teacher approaches. Aiming to fill this gap, we frame the multi-teacher representation distillation as a task-enabling problem. Our goal is to create representations that capture as much information as possible, allowing them to be useful for a wide range of tasks, even without prior knowledge on those tasks. We propose guiding the student model to learn representations that, when applied to downstream tasks, produce predictions aligned with the majority of the predictions obtained from the teachers' representations. This strategy enables our method to harness the collective knowledge of the teachers' ensemble.

For a given task, we formally introduce an ensembling loss that measures the agreement of the Bayesian predictor using the student's embeddings and the Bayesian predictors using the teachers'. We then show that it can be bounded independently of the task by the conditional entropy of the teachers' embedding, knowing the student's output, providing a task-agnostic student-teachers reconstruction loss.

**Contributions.** Our contributions are threefold:

1. **A task-enabling distillation setting.** We frame the multi-teacher distillation problem in a task-enabling setting, in which we study the relationship between the Bayes classifiers obtained from the students and the teachers' embeddings. We show that the conditional entropy of the teachers given the student's output controls the probability of the student's Bayesian predictor disagreeing with the teachers' for any task.
2. **A practical implementation.** We leverage a recent estimator of the differential conditional entropy in high dimension to build an end-to-end optimization framework to minimize our task-agnostic loss.
3. **High-quality embedders.** We demonstrate that our method enhances distillation capabilities across three application domains: computer vision, molecular modeling, and natural language processing, and release trained students that achieve high performance on a diverse range of tasks.

## 2 RELATED WORK

**Task-oriented Distillation.** Knowledge Distillation (KD) is widely used for transferring knowledge from one or a set of teachers to a student model (Gou et al., 2021) in order to improve the performance of the student on a given task (Zhang et al., 2019; Yim et al., 2017). This is typically done by transferring logits (Sun et al., 2024); *i.e.* the models' output, features (Wang et al., 2023; Sarkar & Etemad, 2024), relational information (Dong et al., 2024; 2021), or a mixture of them (Liu et al., 2021a). Similarly, (Qiu et al., 2024) use a regularization term to distill the task-relevant information from the large teacher to the small student. We depart from these methods by focusing on distilling task-agnostic representations.

**Task oriented Multi-Teacher Distillation.** A common method for multi-teacher knowledge distillation is averaging the teachers' logits and transferring the result to the student (Dvornik et al., 2019; Hinton et al., 2015). However, this approach is not ideal when the performance of the teachers is uncertain. Alternative methods include using gate networks (Zhu et al., 2020), reinforcement learning agents (Yuan et al., 2020), and other methods (Ma et al., 2024a; Borza et al., 2022; Zhang et al., 2023) to perform teacher selection or evaluation. Due to challenges in distilling knowledge among diverse architectures, multi-teacher knowledge distillation research mainly focuses on logit distillation. For feature distillation, mean squared error (MSE) is the primary loss function (Gong & Wen, 2024; Navaneet et al., 2022). Other techniques were also explored, such as multi-teacher feature ensemble (Ye et al., 2024), contrasting feature distillation (Li et al., 2024), and cosine similarity-based methods for various tasks (Ma et al., 2024b; Aslam et al., 2024; 2023). Although successful, most multi-teacher feature distillation methods remain oriented to only one or a few set of tasks. These methods are also mostly applicable among teachers and students with different architectures only with the help of an auxiliary classifier (Yang et al., 2021).

**Task-agnostic features and representations distillation.** To the best of our knowledge, few works address task-agnostic representation distillation, and none in a multi-teacher setting. Some works induce strong limitations, such as requiring the student and teachers to have the same architecture (Liang et al., 2023; Xu et al., 2022b), or by requiring to fine-tune the teachers to then distill their representations (Liu et al., 2023). Some other work induce less requirements, notably Gao et al. (2022) rely on vision specific data augmentation, RoB (Duval et al., 2023) focuses on the distillation of joint-embedding approaches, and SEED (Fang et al., 2021) imposes both the student's and the teacher's embeddings to have the same dimension. Finally, Abbasi Koohpayegani et al. (2020) proposed a method ("1-q") with almost no requirements on the student's architecture, measuring the similarity between different embeddings to obtain logits and minimize the KL-divergence between the student's and the teacher's logits. However, all of these methods focus only on the single teacher setting. A related line of work to build more informative representations is contrastive learning (Feng

et al., 2024; Liu et al., 2022; Xu et al., 2022a). However, these methods jointly train the student and the teachers or necessitate defining positive and negative pairs, which is not trivial in some domains.

**Interval estimation.** Most works in distillation rely on MSE or Cosine base distillation, effectively using point estimation methods. However, it is well known in Reinforcement Learning that these standard regression methods are difficult to train (Farebrother et al., 2024). On the other hand, replacing traditional regression scheme by maximum-likelihood training of Gaussian kernels is more stable (Stewart et al., 2023) and effective in Value learning (Bellemare et al., 2017). We extend this idea in the context of embedder distillation by using Gaussian kernels to estimate the conditional distribution of the teachers' embeddings given the student embedding and show that it is directly connected to maximizing the mutual information between the student and the teacher.

## 3 DISTILLING REPRESENTATION THROUGH GAUSSIAN KERNELS

### 3.1 BACKGROUND & NOTATIONS

We suppose that every space $\mathcal{X}$ is a standard Borel Crauel (2002), equipped with its Borel $\sigma$-algebra $\mathcal{B}(\mathcal{X})$. We denote by $\mathbf{X}$ any random variable taking its value onto a space $\mathcal{X}$, and by $\mathcal{P}(\mathcal{X})$, the set of all probability measures over $\mathcal{X}$. $P_{\mathbf{X}} \in \mathcal{P}(\mathcal{X})$ will refer to the induced distribution of $\mathbf{X}$ over $\mathcal{X}$ (push-forward measure). For $P_{\mathbf{X}} \in \mathcal{P}(\mathcal{X})$, we suppose the existence of its density function $f_{\mathbf{X}}$.

**Setting.** In the following, $\mathcal{X}$, will refer to the input space (data) and $\mathbf{X} \sim P_{\mathbf{X}}$ to the input distribution. We suppose we have access to a dataset $\mathcal{D} = \{\mathbf{x}_i\} \subset \mathcal{X}$ of inputs, i.i.d accordingly to $P_{\mathbf{X}}$, and different teacher embedders $\mathsf{T}_k : \mathcal{X} \to \mathbb{R}^{d_k}$, $k \in \{1, \ldots, K\}$, that map the inputs to different embedding spaces.

**Conditional Differential Entropy (Cover & Thomas, 2006).** For a random variable $\mathbf{U}$, defined on $\mathcal{U}$, the differential entropy of its distribution is defined as: $h(\mathbf{U}) = -\int_{\mathcal{U}} f_{\mathbf{U}}(u) \log f_{\mathbf{U}}(u) du$, where $f$ is the probability density of $\mathbf{U}$. For two random variables $\mathbf{U}$ and $\mathbf{V}$, taking their values on $\mathcal{U}$ and $\mathcal{V}$ respectively, the conditional differential entropy of $\mathbf{U}$ given $\mathbf{V}$ is defined as:

$$h(\mathbf{U}|\mathbf{V}) = -\int_{\mathcal{U} \times \mathcal{V}} F_{\mathbf{UV}}(du, dv) \log f_{\mathbf{U}|\mathbf{V}}(u|v).$$

This quantity measures how predictable is $\mathbf{U}$ given the value the observation $\mathbf{V}$. If the two random variables are independent, then the conditional differential entropy is equal to the differential entropy of $\mathbf{U}$, in other words, knowing $\mathbf{V}$ does not provide any information about $\mathbf{U}$.

### 3.2 FROM A TASK-ORIENTED SETTING TO A TASK-AGNOSTIC LOSS

Our goal is to train a representation model capable of effectively handling any downstream task, by leveraging diverse representations from diverse pretrained teachers. To do so, we first measure the agreement between the student's Bayes classifier and the teachers' for any given task. We show that it can be bounded by the conditional entropy of the teacher's embedding given the student's, which does not depend on the considered task.

Let us consider a task characterized by a target set $\mathcal{Y}$ of discrete concepts and the feature space $\mathcal{X}$ with joint probability measure $P_{\mathbf{YX}} \in \mathcal{P}(\mathcal{Y} \times \mathcal{X})$ induced by random variables $(\mathbf{Y}, \mathbf{X}) \in \mathcal{Y} \times \mathcal{X}$. For every projection of the features through the different teachers, we can define the Bayes decision rule $c_{\mathsf{T}_k}^* \triangleq \arg\max_{c:\mathbb{R}^{d_k} \to \mathcal{Y}} \mathbb{E}_{\mathbf{XY}}\left[\mathbb{1}\left[c(\mathsf{T}_k(\mathbf{X})) = \mathbf{Y}\right]\right]$ and similarly for the student: $c_{\mathsf{S}}^* \triangleq \arg\max_{c:\mathbb{R}^d \to \mathcal{Y}} \mathbb{E}_{\mathbf{XY}}\left[\mathbb{1}[c(\mathsf{S}(\mathbf{X})) = \mathbf{Y}]\right]$.

Our goal is to minimize the probability that the student's Bayesian classifier behaves in a different way than the teachers' on each sample. This is shown to improve the performance in most of the cases by decreasing the bias and variance of models and increasing their robustness and generalizability (Dietterich, 2000; Scimeca et al., 2023; Allen-Zhu & Li, 2020; Theisen et al., 2024). In other words, we want to minimize the probability of the student making a different decision than each

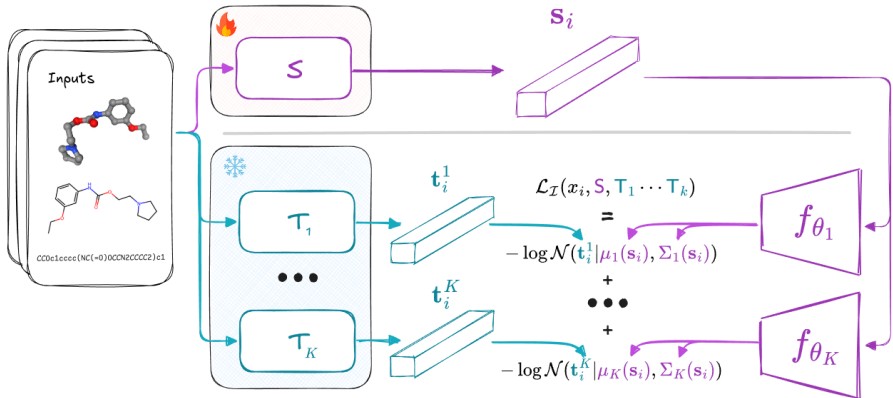

Figure 1: We train our embedder in an end-to-end fashion: we update both the weights of the embedder and that of the Gaussian kernel ($f_\theta$) to minimize the negative log likelihood of the teachers' embedding, given the student output.

teacher:

$$\mathcal{L}^*(\mathbf{Y}, \mathsf{S}, \mathsf{T}_1, \ldots, \mathsf{T}_K) = \frac{1}{K} \sum_{k=1}^{K} \underbrace{\Pr\left(c_\mathsf{S}^*(\mathsf{S}(\mathbf{X})) \neq c_{\mathsf{T}_k}^*(\mathsf{T}_k(\mathbf{X}))\right)}_{\substack{\text{Probability that the student Bayesian classifier's} \\ \text{output is different from the } k^{\text{th}} \text{ teacher's}}} . \tag{1}$$

Where the loss depends on the label's distribution $\mathbf{Y}$, through the definition of the Bayesian classifiers.

We leverage previous results on the performance of the Bayes classifiers from Darrin et al. (2024) to bound the probability of getting a different outcome using Bayes classifiers operating on different projections of the input space.

**Proposition 1** (Darrin et al. (2024)). *Let $C_{\mathsf{T}_k} = c_{\mathsf{T}_k}^*(\mathsf{T}_k(\mathbf{X}))$ and $C_\mathsf{S} = c_\mathsf{S}^*(\mathsf{S}(\mathbf{X}))$ denote the outcome of the Bayes classifier observing the output of the teacher $\mathsf{T}_k$ and the student $\mathsf{S}$, respectively.*

$$\Pr\left(C_\mathsf{S} \neq C_{\mathsf{T}_k}\right) \leqslant 1 - \exp\left(-h(\mathsf{T}_k(\mathbf{X})|\mathsf{S}(\mathbf{X}))\right). \tag{2}$$

**Corollary 1** (Upper bound). *By applying Prop. 1 to Eq. 1, for any target set $\mathcal{Y}$, and label distribution $P_\mathbf{Y}$, we obtain the following bound:*

$$\mathcal{L}^*(\mathbf{Y}, \mathsf{S}, \mathsf{T}_1, \ldots, \mathsf{T}_K) \leqslant 1 - \exp\left(-\underbrace{\frac{1}{K} \sum_{k=1}^{K} h(\mathsf{T}_k(\mathbf{X})|\mathsf{S}(\mathbf{X}))}_{\text{Negative log likelihood}}\right). \tag{3}$$

*The proof of this corollary is straightforward and relies on the concavity of $t \to 1 - \exp(-t)$ (see Appendix A).*

**This bound does not depend on the specific task, but only on the conditional entropy of the teacher embeddings given the student embeddings.** Thus, optimizing the student to minimize this loss provides a task-agnostic approach to aligning the student's Bayes classifier predictions with the ensemble of teachers' predictions across any downstream task.

### 3.3 METHOD

**Estimation of the conditional entropy.** To evaluate the conditional entropy of the teachers' embeddings given the student's, we need a kernel to learn their conditional distribution $\hat{p}(\mathsf{T}_k(\mathbf{X})|\mathsf{S}(\mathbf{X}))$. To estimate this distribution, we use a parametric Gaussian model whose parameters $\mu_k(\mathsf{S}(\mathbf{X}))$ and $\Sigma_k(\mathsf{S}(\mathbf{X}))$ are learned during the training of the student (Pichler et al., 2022).

**Loss function.** Following the above reasoning, we propose to train the student embedder S by simply minimizing the negative log-likelihood (estimated using Gaussian Kernels) of the teachers given the student.

$$\hat{\mathcal{L}}(\mathsf{S}, \mathsf{T}_1, \ldots, \mathsf{T}_K) = \frac{1}{K} \sum_{k=1}^{K} h(\mathsf{T}_k(\mathbf{X}) | \mathsf{S}(\mathbf{X})) \tag{4}$$

$$\approx \frac{1}{K} \sum_{k=1}^{K} \mathbb{E}_{\mathbf{X}} \left[ -\log \mathcal{N}(\mathsf{T}_k(\mathbf{X}) | \mu_k(\mathsf{S}(\mathbf{X})), \Sigma_k(\mathsf{S}(\mathbf{X}))) \right]. \tag{5}$$

Where $\mathcal{N}(\cdot | \mu, \Sigma)$ is the Gaussian distribution with mean $\mu$ and covariance $\Sigma$. In our setting, minimizing the conditional entropy $h(\mathsf{T}_k(\mathbf{X}) | \mathsf{S}(\mathbf{X}))$, exactly corresponds to maximizing the mutual information $I(\mathsf{T}_k(\mathbf{X}); \mathsf{S}(\mathbf{X})) = h(\mathsf{T}_k(\mathbf{X})) - h(\mathsf{T}_k(\mathbf{X}) | \mathsf{S}(\mathbf{X}))$ since for each teacher $h(\mathsf{T}_k(\mathbf{X}))$ is constant w.r.t of the student. This also applies to the bound in Eq. 3.

**Training procedure.** We train both the student and the different kernels in an end-to-end fashion by minimizing the loss function $\mathcal{L}$. It boils down to minimizing the negative log-likelihood of the teachers' embeddings given the student's embedding. We use the Adam optimizer to minimize the loss function. See Appendix F for the detailed training algorithm.

**Baselines and Evaluation.** We consider two mainly used multi-teacher feature distillation methods, MSE and Cosine similarity (see Appendix G for more information). To evaluate the representations learned by the student, for each modality, we run different benchmarks evaluating its performance on a wide variety of downstream tasks. For classification and regression tasks, we train a small feedforward network on top of the embeddings (the backbones are considered frozen) on different tasks and evaluate its performance.

## 4 VISION

### 4.1 EXPERIMENTAL SETTING

Table 1: Comparison of teacher and student models' accuracy on vision modality's tasks with or without different distillation methods (our method (NLL), MSE (L2), Cosine).

| Method | Model | CIFAR10 | FMNIST | MNIST | STL10 | SVHN | QMNIST | KMNIST | CelebA |
|--------|-------|---------|--------|-------|-------|------|--------|--------|--------|
| | resnet18 | 81.89 | 86.94 | 96.6 | 92.98 | 51.01 | 96.89 | 80.43 | 90.82 |
| | squeezenet | 79.23 | 86.65 | 97.51 | 85.82 | 47.77 | 97.59 | 84.05 | 61.35 |
| | densenet | _87.49_ | 88.69 | 96.80 | **97.11** | 66.91 | 97.72 | 86.33 | 93.98 |
| | googlenet | 81.94 | 86.38 | 96.71 | 93.95 | 55.9 | 97.2 | 79.27 | 92.93 |
| NoKD | shufflenet | 81.61 | 87.57 | 95.77 | 71.51 | 49.08 | 95.96 | 76.97 | 92.42 |
| | mobilenet | 81.67 | 88.07 | 96.05 | 92.26 | 48.57 | 97.5 | 85.64 | 91.02 |
| | mnasnet | 81.41 | 88.76 | 96.09 | 92.79 | 57.63 | 97.00 | 82.35 | 89.01 |
| | resnext50-32x4d | 83.42 | 87.32 | 95.37 | _95.97_ | 52.87 | 96.65 | 83.37 | 91.74 |
| | wide-resnet50-2 | 84.30 | 87.40 | 95.16 | 95.85 | 57.77 | 96.74 | 76.23 | 90.22 |
| Cosine | resnet18 | 84.57 | _89.90_ | 98.58 | 88.34 | _76.34_ | 98.95 | _91.97_ | _95.00_ |
| L2 | resnet18 | 82.90 | 89.75 | 98.25 | 88.15 | 74.84 | 98.61 | 88.21 | 94.89 |
| NLL | resnet18 | **87.51** | **90.64** | **99.15** | 88.45 | **81.99** | **99.15** | **95.21** | **95.47** |

**Teachers and evaluations.** We gather general models from available models of Torchvision, including ResNet18 (He et al., 2016), ResNext (Xie et al., 2017), WideResNet (Zagoruyko & Komodakis, 2017), SqueezeNet (Iandola et al., 2016), DenseNet (Huang et al., 2017), GoogLeNet (Szegedy et al., 2015), ShuffleNet (Ma et al., 2018), MobileNet (Sandler et al., 2018), and MNASNet (Tan et al., 2019). For more information about the models, refer to Sec. D.2[1].

---

[1] https://anonymous.4open.science/r/vision-distill-2E6C

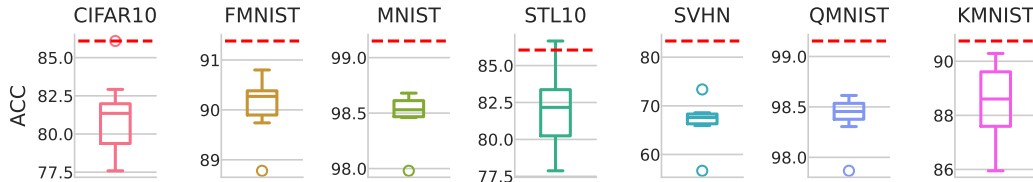

Figure 3: Accuracy comparison between multi-teacher (red line) and single-teacher (box-plots) distillation for all available teachers on each task.

**Training set.** We use the official training set of generic datasets available from Torchvision, including CIFAR10 (Krizhevsky et al., 2009), FashionMNIST (Xiao et al., 2017), MNIST (Deng, 2012), STL10 (Coates et al., 2011), CelebA (Liu et al., 2015), SVHN (Netzer et al., 2011), QMNIST (Yadav & Bottou, 2019), and KMNIST (Clanuwat et al., 2018). For more information, refer to Sec. D.1.

### 4.2 RESULTS

**Comparison with different multi-teacher feature distillation methods.** We compare the downstream performance of each embedder, with that of the student models. For all experiments, ResNet18 is considered as the student backbone and trained using our distillation method. While our experimental setting (freezing the backbones and training a feedforward on the embeddings for each task) leads to weaker performances overall, it enables us to effectively compare the quality of the embeddings generated.

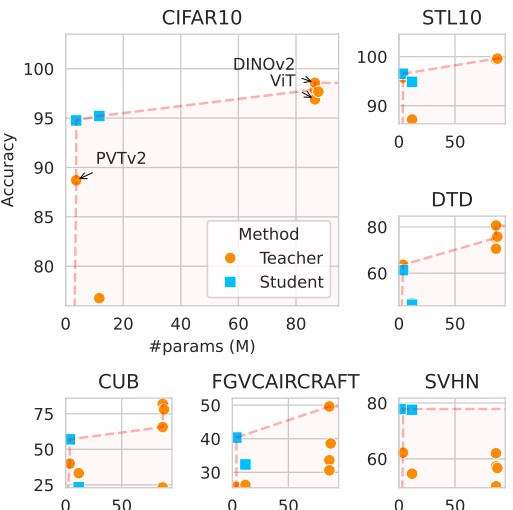

Figure 2: Pareto frontier of vision models, showing that models distilled using our method (blue) sit on the Pareto frontier.

Tab. 1 shows the accuracy of teachers and the student with different distillation methods per task, demonstrating that our method outperforms others in terms of accuracy in all cases, but one (STL 10). Detailed results for other student architectures can be found in Sec. D.3.

**Comparison with Single-teacher distillation.** Finally, we trained the student using our approach in a single-teacher setting to evaluate how much incorporating multiple teachers improves the quality of the learned representations[2]. Figure 3 displays the accuracy of the student distilled from different single teachers, compared to the multi-teacher scenario. On all tasks, using multiple teachers improves the performances of the student model, with the only exception of STL 10, where the student trained with only densenet slightly outperforms our multi-teacher baseline. For the detailed results, refer to Sec. D.3.

**Vision Transformer Experiments** To further evaluate our method, we experiment with Vision Transformer teachers (Swin (Liu et al., 2021b), DINOv2 (Oquab et al., 2023), ViT (Dosovitskiy et al., 2021), BEiT (Bao et al., 2022) and PVTv2 (Wang et al., 2022b)). For our students, we use ResNet18 ( 12M parameters) and PVTv2 ( 4M parameters), a relatively smaller Vision Transformer. We performed our evaluation on DTD (Cimpoi et al., 2014), FGVCAircraft Maji et al. (2013), and CUB (Welinder et al., 2010), in addition to CIFAR10, SVHN, STL10. As shown in Figure 2, the distilled students achieve the best results for models of their size, except for DTD, where the original version of PVTv2 slightly outperforms our ViT student. Detailed results can be found in Sec. D.3.

---

[2]All students were initialized with ResNet18

Table 2: Average rank of each model on the ADMET and HTS downstream tasks from the TDC (Huang et al., 2021) platform. Our student outperform all baselines including teachers on average.

| | Absorption | Distribution | Metabolism | Excretion | Tox | HTS | Avg |
|---|---|---|---|---|---|---|---|
| InfoGraph | 13.50 | 13.27 | 13.32 | 11.40 | 11.98 | 9.40 | 12.14 |
| ChemBertMLM-10M | 10.65 | 11.00 | 10.70 | 13.80 | 11.11 | 14.60 | 11.98 |
| FRAD QM9$^{(t)}$ | 10.57 | 11.13 | 10.38 | 8.33 | 10.04 | 7.80 | 9.71 |
| ChemGPT-1.2B | 9.55 | 11.73 | 11.75 | 10.73 | 10.86 | 11.20 | 10.97 |
| GROVER | 10.43 | 8.33 | 11.25 | 8.53 | 10.38 | 11.00 | 9.99 |
| GraphCL$^{(t)}$ | 10.89 | 8.53 | 9.45 | 10.13 | 8.70 | 9.80 | 9.58 |
| GraphLog$^{(t)}$ | 11.05 | 7.80 | 9.07 | 10.53 | 8.93 | 14.00 | 10.23 |
| GraphMVP$^{(t)}$ | 7.20 | 6.20 | 7.85 | 9.80 | 7.49 | 8.80 | 7.89 |
| MolR gat | 6.95 | 7.60 | 8.30 | 8.53 | 6.49 | _3.40_ | 6.88 |
| ThreeDInfomax$^{(t)}$ | _4.17_ | _6.00_ | 7.58 | 7.13 | 6.16 | 10.40 | 6.91 |
| ChemBertMTR-77M$^{(t)}$ | **3.50** | **4.27** | 5.75 | 5.00 | 6.03 | 4.20 | _4.79_ |
| L2 | 8.07 | 6.40 | 5.55 | 6.33 | 7.55 | **3.00** | 6.15 |
| Cosine | 5.51 | 6.13 | _3.60_ | _4.33_ | **4.97** | 6.20 | 5.13 |
| student-250k | **3.55** | 6.20 | **2.70** | **2.40** | _4.99_ | 3.80 | **3.94** |
| student-2M | 4.40 | **5.40** | 2.75 | 3.00 | _4.34_ | 2.40 | _3.72_ |

# 5 MOLECULAR MODELING

## 5.1 EXPERIMENTAL SETTING

**Teachers and Architecture** We use 9-teachers trained on different modalities: SMILES (textual representation of the molecular graph) (Ahmad et al., 2022), 2D molecular graphs (You et al., 2020; Xu et al., 2021; Liu et al., 2022; Stärk et al., 2021), and 3D structures (Zaidi et al., 2023; Feng et al., 2023). We identify the teachers with: $^{(t)}$ such as ChemBERTaMTR$^{(t)}$, and use a 2D-GNN (Graph Isomorphism Network: GIN (Hu et al., 2020)) for our student (for more details see Sec. B.1)[3].

**Evaluation setting** We evaluated all models on the ADMET (Absorption, Distribution, Metabolism, Excretion, Toxicity) tasks of the Therapeutic Data Commons platform (TDC) (Huang et al., 2021) and on high throughput screening task (HTS), (HIV (Wu et al., 2018)). We record the test performance over 5 runs (details on the evaluation procedure in Sec. B.3).

**Dataset** We trained our models on two datasets: the ZINC-250k (Irwin & Shoichet, 2005), consisting of 250,000 samples, and a processed version of the ZINC Clean Leads dataset (Polykovskiy et al., 2018), containing 2 million samples. Both are public datasets of commercially available compounds, designed to be used in various therapeutic projects.

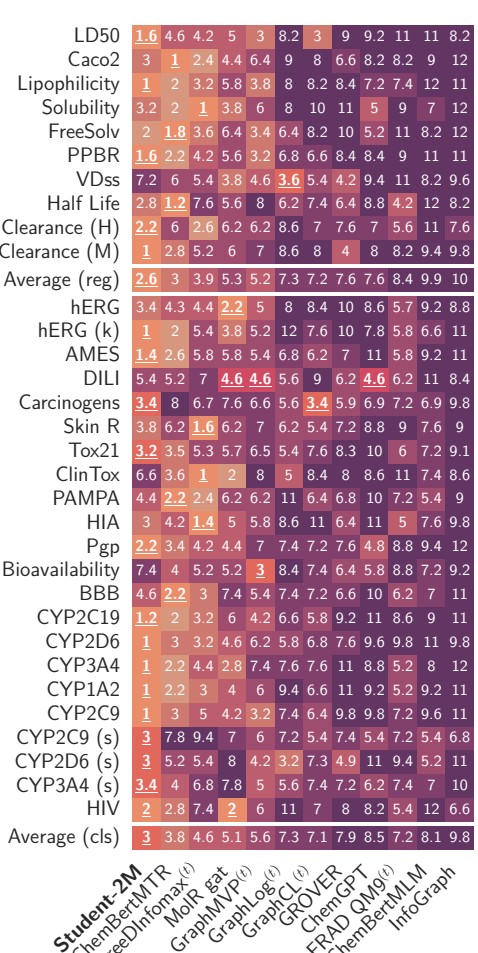

Figure 4: Rank of each model on molecular regression and classification tasks.

[3]https://anonymous.4open.science/r/mol-distill-DE87

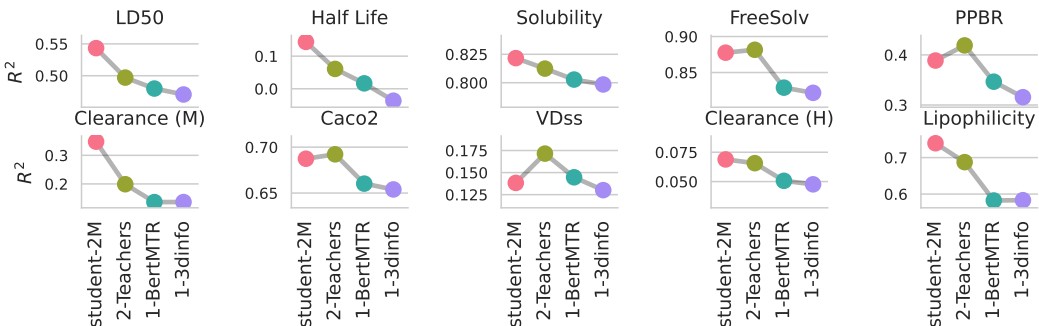

Figure 5: Test $R^2$ score of the students on the regression tasks, trained with all teachers, two teachers and one teacher ("1-BertMTR" for ChemBertMTR and "1-3dinfo" for 3D-infomax).

## 5.2 RESULTS

**Overall performance.** We compare the performance of the student model with the teachers and other baseline embedders on the different tasks. We report the mean rank of every model on each category of tasks in Tab. 2. On average, our student model outperforms all other baselines, achieving consistent competitive results in every task category.

**Consistent performance.** Results (average rank) for each task are presented in Figure 4. Our student model achieves the best performance on both regression and classification tasks, delivering the most accurate predictions across a majority of tasks. This suggests that our method generates informative representations thereby providing high-quality molecular descriptors.

**Dataset size impact.** Surprisingly, the performances of the "student-250k" and "student-2M" models are similar on average. Specifically, the student-250k model outperforms the student-2M model on regression datasets notably, by achieving the best performances on the FreeSolv (Mobley & Guthrie, 2014) and Lipophilicity (Wenlock & Tomkinson, 2021) tasks. This suggests that our method can leverage the diversity of the teachers to learn more informative representations, even when trained on a smaller dataset of 250k datapoints.

**Single teacher vs. Multi-Teachers.** To assess the impact of training a student on multiple teachers, we trained students to distill the knowledge of a single teacher, and two teachers. We chose the two of the best performing baselines as teachers, ChemBERTaMTR-77M (Ahmad et al., 2022) and 3D-infomax (Stärk et al., 2021), and trained the student model on the 2M-molecules dataset. Figure 5 displays the results on regression tasks (further details in Sec. B.4). The students trained with a single teacher are outperformed by the student trained with the two teachers. Besides, the student-2M trained with all teachers outperforms all these students. Training with multiple teachers thus appears to be beneficial as it allows it to learn more informative representations.

## 6 NATURAL LANGUAGE PROCESSING DISTILLATION

We apply our method to text embedders in a slightly different setting than molecular representations and vision: we focus on distilling strong and large models into significantly smaller ones. Indeed, modern models in NLP are extremely large and costly to train[4]. Thus, we aim to produce the best possible models for a given weight, pushing the size/performance of the Pareto frontier (Figure 6), and not necessarily competing with the largest models. We trained and evaluated models of 3 different sizes (22M, 33M and 109M) based on the snowflakes Merrick et al. (2024) embedders. We release SOTA models for classification and clustering tasks.

## 6.1 EXPERIMENTAL SETTING

---

[4]https://anonymous.4open.science/r/NLP-MultiTeacherDistillation

Table 3: Performance of our distilled models compared to the best models of similar sizes from the MTEB Benchmark on classification tasks.

| | Task
Model | Size | Amazon Counterfactual | Amazon Polarity | Amazon Reviews | Banking77 | Emotion | Imdb | MTOPDomain | MTOPIntent | Massive Intent | Massive Scenario | Toxic Conversations | Tweet Sentiment Extraction | Avg. |
|---|---|---|---|---|---|---|---|---|---|---|---|---|---|---|---|
| **xs** | MTEB | GIST | 23M | 72.9 | **87.2** | 42.6 | 84.2 | 52.1 | 78.5 | 94.8 | 77.7 | 73.2 | 76.7 | **72.9** | 59.9 | 72.7 |
| | | Ivysaur | 23M | 72.1 | 86.7 | **42.7** | 81.9 | 45.4 | 80.8 | 92.1 | 71.9 | 70.3 | 74.9 | 65.5 | 58.7 | 70.2 |
| | | gte-tiny | 23M | 71.8 | 86.6 | 42.6 | 81.7 | 44.7 | 80.5 | 91.8 | 69.9 | 70.1 | 74.9 | 71.0 | 58.6 | 70.3 |
| | MSE | Student-xs | 23M | 71.6 | 86.2 | 42.3 | 83.6 | 57.5 | **83.5** | 94.5 | 75.4 | 74.3 | **80.4** | 66.3 | 59.3 | 72.9 |
| | NLL | Student-xs | 23M | **76.5** | 84.9 | 42.4 | **85.8** | **58.0** | 81.1 | **95.2** | 79.9 | 75.8 | 80.4 | 68.1 | **60.1** | **74.0** |
| **s** | MTEB | bge-small-en-v1.5 | 33M | 73.8 | 92.8 | 47.0 | 85.7 | 47.8 | **90.6** | 93.4 | 74.8 | 74.8 | 78.7 | 69.9 | 60.5 | 74.1 |
| | | GIST | 33M | 75.3 | 93.2 | 49.7 | 86.7 | 55.9 | 89.5 | 95.5 | 79.1 | 75.5 | 79.2 | **72.8** | 61.0 | **76.1** |
| | | NoInstruct | 33M | 75.8 | **93.3** | **50.0** | 86.4 | 55.1 | 90.2 | 95.3 | 79.6 | 76.0 | 79.3 | 69.4 | **61.3** | 76.0 |
| | MSE | Student-s | 33M | 72.6 | 90.3 | 44.3 | 84.2 | 56.5 | 88.8 | 94.9 | 77.2 | 75.4 | **81.2** | 64.9 | 60.4 | 74.2 |
| | NLL | Student-s | 33M | **77.3** | 89.2 | 43.8 | **86.7** | **58.0** | 88.3 | 95.5 | 81.9 | 76.7 | 80.7 | 66.1 | 60.6 | 75.4 |
| **m** | MTEB | bge-base-en-v1.5 | 109M | 76.2 | 93.4 | 48.9 | 87.0 | 51.9 | **90.8** | 94.2 | 76.9 | 76.2 | 80.2 | 71.6 | 59.4 | 75.5 |
| | | GIST | 109M | 76.0 | **93.5** | **50.5** | 87.3 | 54.7 | 89.7 | 95.3 | 78.1 | 76.0 | 79.6 | **72.4** | 59.3 | 76.0 |
| | | e5-base-4k | 112M | **77.8** | 92.8 | 46.7 | 83.5 | 47.0 | 86.2 | 93.7 | 75.3 | 73.0 | 77.7 | 72.1 | 60.4 | 73.8 |
| | | e5-base-v2 | 110M | **77.8** | 92.8 | 46.7 | 83.5 | 47.0 | 86.2 | 93.7 | 75.3 | 73.0 | 77.7 | 72.1 | 60.4 | 73.8 |
| | MSE | Student-m | 109M | 76.6 | 89.1 | 44.7 | 87.2 | **60.8** | 88.0 | 95.7 | 81.6 | 77.7 | 82.2 | 67.3 | 60.5 | 76.0 |
| | NLL | Student-m | 109M | **79.6** | 89.5 | 45.8 | **88.0** | 59.7 | 88.3 | **96.2** | 83.9 | 78.6 | 82.7 | 67.1 | **61.3** | **76.7** |

**Teachers and student.** We select four freely available embedding models from the Huggingface hub (Wolf et al., 2020) (See Sec. C.1.2 for a detailed list of the teachers) whose evaluations are available in the MTEB benchmark (Muennighoff et al., 2023). To ensure having a point of comparison, we select teachers of different sizes and performances. Notably, SFR-Embeddings-R_2 is more than ten points stronger than the other three (smaller) teachers.

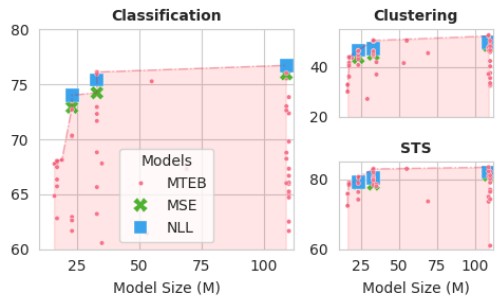

Figure 6: Pareto frontier in NLP. The models distilled using our method (blue) sit on the Pareto frontier.

**Embedder evaluation.** Evaluating NLP models is notably challenging, and the common practice of evaluating a model using multi-task benchmarks may not be indicative of model capabilities (Liu et al., 2024). For lack of better options and because it is currently the most widely accepted benchmark, we rely on the evaluation provided by the MTEB benchmark (Muennighoff et al., 2023) for clustering, sentence similarities and classification tasks.

**Training set.** We gathered different common datasets used for training embedders and collected 6 million entries from the Huggingface Hub, including Specter Cohan et al. (2020), T5 Ni et al. (2021), Amazaon QA McAuley & Leskovec (2013), IMDB Maas et al. (2011), SNLI Bowman et al. (2015), QQP triplets from Quora, AG News Zhang et al. (2015), MEDI dataset Su et al. (2023) and the DAIL Emotion dataset Saravia et al. (2018). We provide the dataset statistics in Sec. C.1.1. The datasets are all flattened, such that if the original had two columns (e.g., sentence 1 and sentence 2 in the SNLI dataset), we end up with twice the number of entries, one for each sentence, and we deduplicated the dataset.

**Model Architecture.** We use as starting points the snowflakes Merrick (2024); Merrick et al. (2024) models xs (22M), s (33M) and m (109M) and we further train them using our distillation method (See Sec. C.1.4).

## 6.2 DISTILLATION PERFORMANCE

**Task performance.** Our method produces models that exhibit strong performance on a large variety of tasks, ranking first amongst all models of similar size in the MTEB benchmark on most of the tasks (See Figure 7). Notably, we observe that our method produces models that are competitive for almost all the tasks, whereas other models appear more specialized. We provide the actual accuracy of our models on classification tasks in Tab. 3. We provide the full results for all model sizes in Sec. C.2.1.

**Pareto frontier.** We show in Figure 6 that our method can pack more information into fixed-size models, pushing the Pareto frontier between model size and downstream performance. Our models of 22M, 33M, and 109M parameters all sit on the Pareto frontier, providing new state-of-the-art models in their respective size categories.

**Embedding space structure.** As our metric only optimizes the mutual information between the student and the teachers, it does not directly enforce any structure on the embedding space. Indeed, information is invariant through invertible transformations. Let $f_1$ and $f_2$ be differentiable and invertible mapping function (diffeomorphism), then $I(X;Y) = I(f_1(X); f_2(Y))$. As a result, our objective does not enforce the preservation of the teachers' embedding space structural properties (such as pairwise cosine similarity). Surprisingly, we found that while our method does not provide structural guarantees over the embedding space of the student, it was able to retain competitive performance in both clustering (Figure 7 and Figure 6) and STS tasks. For example, on clustering tasks, our largest model (109M) reaches an average V-measure of 50 while the best model achieves 53, and most models of similar sizes fall below 45. Similarly, our models remain on par with the SOTA models in STS tasks (82.1 against 83.5 spearmann correlation). These results are consistent across all three model sizes (See Sec. C.2 for full results).

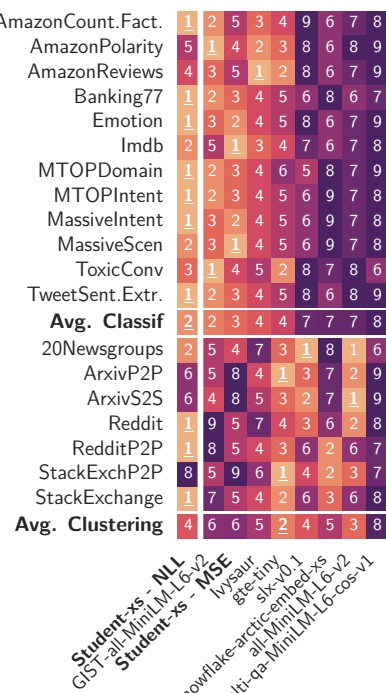

Figure 7: Global ranking on clustering and classification tasks for our medium-sized model (109M)

## 7 CONCLUSIONS AND FUTURE WORK

We proposed a theoretically grounded task-agnostic distillation mechanism that leverages interval estimation through Gaussian kernels in high dimensions to distill a more informative representation from multiple teachers to a single student. We show theoretically that our method maximizes the mutual information and reconstructive power of the student to the teachers and experimentally validate that our method is, in fact, more stable and efficient than point estimation-based multi-teacher feature distillation methods such as MSE or cosine-based distillation mechanisms.

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

# Appendix

## Table of Contents

## A    THEORETICAL RESULT

We denote $\mathbf{X}$ as the random variable over $\mathcal{X}$ that describes the input distribution. We suppose we have access to a dataset $\mathcal{D} = \{\mathbf{x}_i\} \subset \mathcal{X}$ of inputs drawn following $p_{\mathbf{X}}$ and different embedders $\mathsf{T}_k : \mathcal{X} \to \mathbb{R}^{d_k}$, $k \in \{1, \dots, K\}$, that map the inputs to different embedding spaces. We denote $\mathbf{Z_k} = \mathsf{T}_k(\mathbf{X})$ as the random variable over $\mathbb{R}^{d_k}$ that describes the embedding of the input distribution in the $k$-th embedding space and by $\mathbf{U} = \mathsf{S}(\mathbf{X})$ the random variable over $\mathbb{R}^d$ that describe the embedding of the input distribution in the student embedding space. We denote by $\mathbf{z}_i^k = \mathsf{T}_k(\mathbf{x}_i)$ the embedding of $\mathbf{x}_i$ in the $k$-th embedding space. We are interested in learning a representation that captures the information contained in all the embeddings.

Let us consider any target set $\mathcal{Y}$ of discrete concepts over the feature space $\mathcal{X}$ with joint probability measure $P_{YX} \in \mathcal{P}(\mathcal{Y} \times \mathcal{X})$ induced by random variables $(Y, X) \in \mathcal{Y} \times \mathcal{X}$.

By applying the above proposition to all the terms in Eq. 1, we obtain the following bound on the loss function:

**Corollary 2** (Upper bound).

$$\mathcal{L}^*(\mathbf{Y}, \mathsf{S}, \mathsf{T}_1, \dots, \mathsf{T}_K) \leqslant \frac{1}{K} \sum_{k=1}^{K} \left( 1 - \exp\left( -h(\mathsf{T}_k(X)|\mathsf{S}(X)) \right) \right) \tag{6}$$

$$\leqslant 1 - \exp\left( -\underbrace{\frac{1}{K} \sum_{k=1}^{K} h(\mathsf{T}_k(X)|\mathsf{S}(X))}_{\textit{Negative log likelihood}} \right). \tag{7}$$

*Proof.*

$$\mathcal{L}^*(\mathbf{Y}, \mathsf{S}, \mathsf{T}_1, \dots, \mathsf{T}_K) \leqslant \frac{1}{K} \sum_{k=1}^{K} \left( 1 - \exp\left( -h(\mathsf{T}_k(X)|\mathsf{S}(X)) \right) \right)$$

$$\leqslant 1 - \frac{1}{K} \sum_{k=1}^{K} \exp\left( -h(\mathsf{T}_k(X)|\mathsf{S}(X)) \right)$$

$$\leqslant 1 + \frac{1}{K} \sum_{k=1}^{K} -\exp\left( -h(\mathsf{T}_k(X)|\mathsf{S}(X)) \right)$$

$$\leqslant 1 - \exp\left( -\frac{1}{K} \sum_{k=1}^{K} h(\mathsf{T}_k(X)|\mathsf{S}(X)) \right).$$

We simply rearrange the terms and use the fact that $x \mapsto -\exp(-x)$ is concave to interchange the sum and the exponential. $\square$

## B MOLECULAR MODELLING

### B.1 MODEL ARCHITECTURE

We trained a 10-layer GINE (Hu et al., 2020) neural network with a 512 hidden dimension, using a 2-layer network for the message passing process. We use the atomic number of each node as input, as well as possible chirality information, and the nature of the bond between each pair of nodes. We use a batch size of 256 and a learning rate of $1e-4$ to train the model for 400 epochs on the 250k dataset and 200 epochs on the 2M dataset. For the teacher-specific kernels, we used a 3-layer MLP with a hidden size of 1024.

#### B.1.1 CHOSEN TEACHERS

The teachers used to train our molecular modeling students are summed up in Tab. 4. We gathered various representation models for molecular modeling, with different pre-training objectives, input modalities, architectures, and training datasets.

| Model name | SMILES | 2D-GNN | 3D-GNN | Architecture | Out size | Dataset (size) |
|---|---|---|---|---|---|---|
| GraphCL(You et al., 2020) | | ✓ | | GIN | 300 | GEOM (Axelrod & Gómez-Bombarelli, 2022) (50k) |
| GraphLog(Xu et al., 2021) | | ✓ | | GIN | 300 | GEOM (Axelrod & Gómez-Bombarelli, 2022) (50k) |
| GraphMVP(Liu et al., 2022)[1] | | ✓ | | GIN | 300 | GEOM (Axelrod & Gómez-Bombarelli, 2022) (50k) |
| 3D-infomax(Stärk et al., 2021)[1] | | ✓ | | PNA | 800 | QMugs (Isert et al., 2021) (620k) |
| ChemBERT MTR(Ahmad et al., 2022)[2] | ✓ | | | RoBERTa | 384 | PubChem (Kim et al., 2022) (5M, 10M, 77M) |
| 3D-denosing(Zaidi et al., 2023) | | | ✓ | TorchMD-net | 256 | PCQM4Mv2(Hu et al., 2021) (3.7M) |
| 3D-fractional(Feng et al., 2023) | | | ✓ | TorchMD-net | 256 | PCQM4Mv2(Hu et al., 2021) (3.7M) |

Table 4: Description of all teachers used in our experiments.

#### B.1.2 ARCHITECTURE INFLUENCE

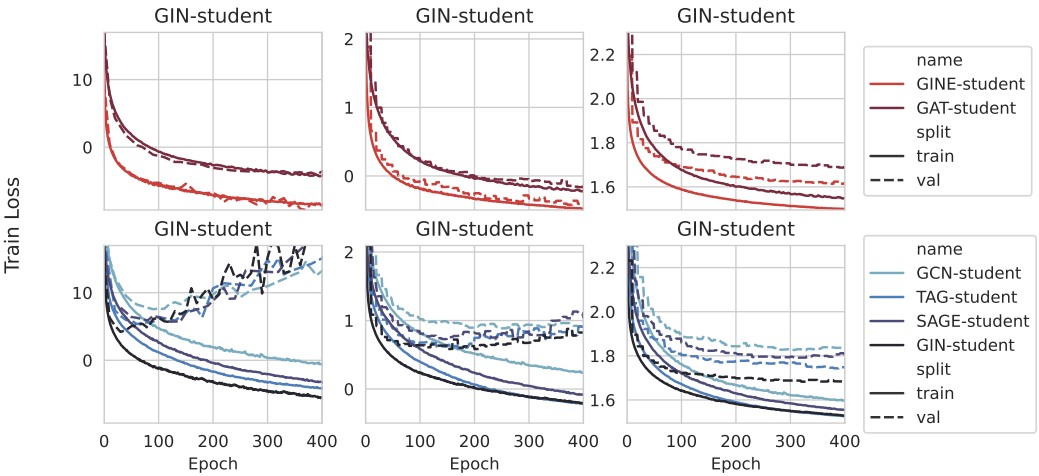

Figure 8: Training loss of different students using different GNN architectures on the ZINC-250k dataset.

Figure 8 shows the training loss of the student model with different GNN architectures on the ZINC-250k dataset. In particular, we compared the GINE architecture with a Graph Convolutional

---

[1]Models aiming at incorporating 3D information into 2D-GNNs models.

[2]We used the three versions of ChemBERT-MTR models trained on 5M, 10M, and 77M.

Network (GCN) (Morris et al., 2021), a Graph Attention Network (GAT) (Brody et al., 2022), a GraphSAGE (SAGE) (Hamilton et al., 2018), a Toplogy Adaptative Graph Convolutional Network (TAG) (Brody et al., 2022), and a GIN Network, that separates from the GINE architecture by the fact that it does not take edge features into account (Xu et al., 2019). We observe that the GINE architectures outperform the other architectures, with a lower training loss, a faster convergence, and a lower validation loss. The Graph attention network (GAT) is the second best performing architecture, but it is still outperformed by the GINE architecture. These two architectures are the only ones to use the edge embeddings in the message passing process, which could explain their better performance.

Indeed, all other architectures perform worse, especially when considering their validation loss computed on 10% of the training set. Specifically, the GIN architecture, not using edge feature, performs significantly worse than the GINE architecture, while having a similar architecture.

For our experiments, we decided to use the GINE architecture, as it performs the best during training and converges faster than the other architectures.

### B.2 KERNEL'S PREDICTIVE POWER

Our method relies on teacher-specific heads to distill the knowledge of each teacher. In this section, we wish to evaluate the impact of the choice of these kernels and their predictive power (in terms of depth) on the performance and training of the student model.

We performed this experiment with kernels of depth 2, 3, and 5, and we trained the student model with these kernels on the ZINC-250k dataset and evaluated the performance of the student model on the ADMET and HTS downstream tasks.

First, during the training, as expected, the more powerful the kernel, the lower the training loss is (see Figure 9), even though the difference is significant, especially between the students using kernels of depth 3 and 5. Overall, the performances of each student on the downstream tasks are similar, underlining the robustness of our method regarding the choice of the kernel's depth (see Figure 10). For our experiments in the main paper, we used a kernel of depth 3, as it enables the best trade-off between computational complexity, and training convergence while providing competitive results on the downstream tasks.

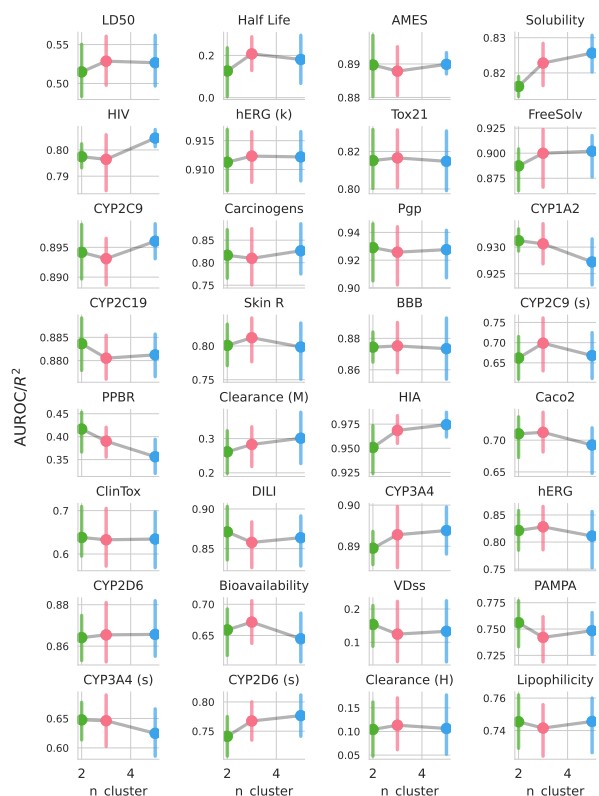

Figure 10: Test AUROC/$R^2$ score of the students on the classification/regression tasks, trained with different kernel-size on the ZINC-250k dataset.

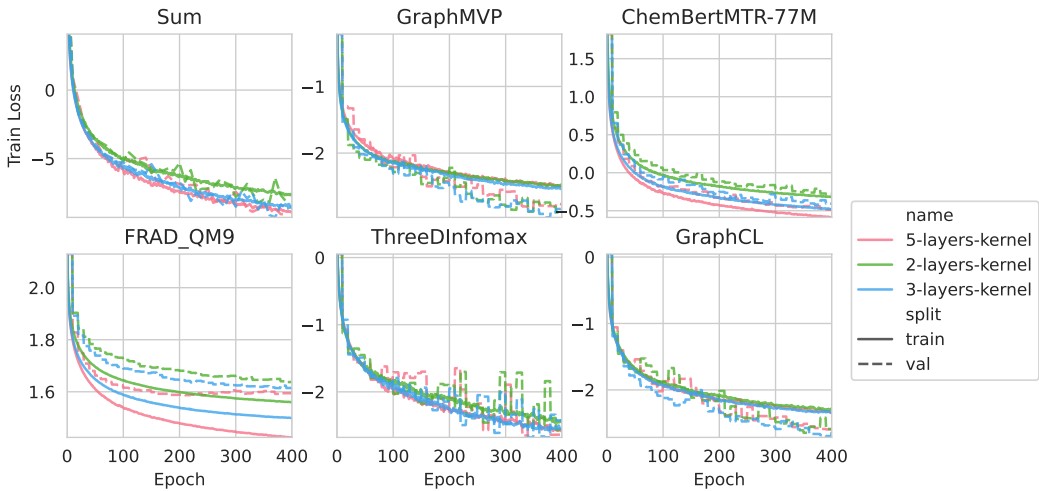

Figure 9: Training loss of the student model along the training with different kernel-size on the ZINC-250k dataset.

### B.3 EVALUATION DETAILS

#### B.3.1 BENCHMARK CHOICE

We selected a total of 32 tasks, extracted from the Therapeutic Data Commons (Huang et al., 2021) platform, 8 absorption tasks, 3 distribution tasks, 8 metabolism tasks, 3 excretion tasks, 9 toxicity tasks and 1 high-throughput screening task. A summary of the tasks considered can be found in Tab. 5, with their corresponding size (total number of samples) and type (classification or regression). For all tasks, we computed 5 conformations for each molecule, and used the least energetic as an input of our 3D models.

#### B.3.2 EVALUATION PROCEDURE

For every task, we opted for a random split since we obtained similar results to a scaffold split, with a faster computation time, with a ratio of 70/10/20 for the train/validation/test sets. For all tasks, we compute the embeddings generated by each model on the task. We then train a 2 layer perceptron with a hidden size of 128 on the task for $\min(100, 200 * \frac{5000}{\text{task size}})$ epochs (to limit the compute time on large tasks) with a learning rate of $1e - 3$. We then select the best checkpoint according to the validation performances and report the test metrics of this checkpoint.

#### B.3.3 EVALUATION METRICS

We repeat this process five times with different seeds in the train-val-test splits in order to enable the establishment of robust rankings using au-

Table 5: Tasks extracted from the Therapeutic Data Commons platform considered in our experiments.

| Category | Model | Task | cls | reg |
|---|---|---|---|---|
| Absorption | P-glycoprotein Inhibition | 1212 | ✓ | |
| | AqSolDB | 9982 | | ✓ |
| | Lipophilicity | 4200 | | ✓ |
| | Caco-2 Permeability | 906 | | ✓ |
| | Human Intestinal Absorption | 578 | ✓ | |
| | FreeSolv | 642 | | ✓ |
| | PAMPA Permeability | 2035 | ✓ | |
| | Oral Bioavailability | 640 | ✓ | |
| Distribution | Plasma-Protein BDR | 1614 | | ✓ |
| | Blood-Brain barrier | 1975 | ✓ | |
| | VDss | 1130 | | ✓ |
| Metabolism | CYPP450 3A4 Inhib. | 12328 | ✓ | |
| | CYPP450 1A2 Inhib. | 12579 | ✓ | |
| | CYPP450 2C19 Inhib. | 12665 | ✓ | |
| | CYPP450 2C9 Inhib. | 12092 | ✓ | |
| | CYPP450 2D6 Inhib. | 13130 | ✓ | |
| | CYPP450 2D6 Substrate | 664 | ✓ | |
| | CYPP450 3A4 Substrate | 667 | ✓ | |
| | CYPP450 2C9 Substrate | 666 | ✓ | |
| Excretion | Clearance hepatocyte | 1020 | | ✓ |
| | Half Life | 667 | | ✓ |
| | Clearance microsome | 1102 | | ✓ |
| Toxicity | Tox21 | 7831 | ✓ | |
| | hERG | 13445 | ✓ | |
| | | 648 | ✓ | |
| | Acute Toxicity LD50 | 7385 | | ✓ |
| | Ames Mutagenicity | 7255 | ✓ | |
| | ClinTox | 1484 | ✓ | |
| | Carcinogens | 278 | ✓ | |
| | Drug Induced Liver Injury | 475 | ✓ | |
| | Skin Reaction | 404 | ✓ | |
| HTS | HIV | 40000 | ✓ | |

torank (Herbold, 2020). We decided to report the ranks of the models to enable the comparison of the models on both classification and regression by simply averaging the rank. To compute the rank on all tasks, we rely on the AUROC score for classification tasks and the $R^2$ score for regression tasks. For the excretion tasks, since the regression labels have a large variance, we decided to apply the regression on the log-values and report the $R^2$ score on the log-values.

### B.4 SINGLE-TEACHER SETTING

To assess the impact of the multi-teacher setting on the performance of the student model, we trained students to distill the knowledge of a single teacher. We used only the two best performing teachers, 3D-infomax (Stärk et al., 2021) and ChemBERTaMTR (Ahmad et al., 2022), to train the student model on the 2M datapoints dataset. We also train a student with both teachers, to see if those two teachers are sufficient to achieve the same performance as the models we presented in the core of the paper.

Figure 11 shows how these students underperform compared to a student trained with all teachers, in terms of AUROC for classification tasks and $R^2$ for regression tasks respectively. These tables also show that the student trained with both teachers performs better than each student trained with only one teacher.

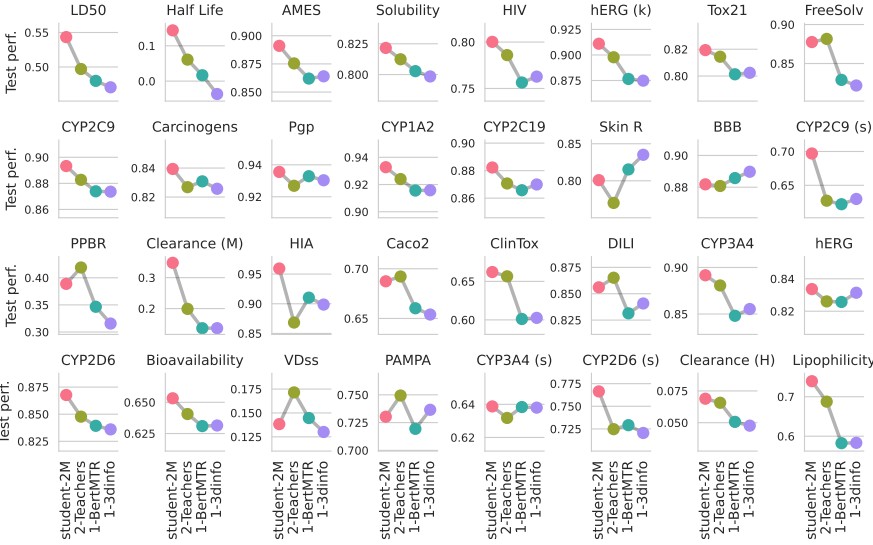

Figure 11: Test AUROC/$R^2$ score of the students on the classification/regression tasks, trained with all teachers (student-2M), two teachers (2-Teachers) and one teacher (1-ChemBertMTR for the model trained with ChemBertMTR-77M and 1-teacher-3dinfomax for the model trained with 3D-infomax).

### B.5 COMPREHENSIVE RESULTS

The following tables provide the raw results of the different evaluated models on the ADMET and HTS downstream tasks. Tab. 6 and Tab. 7 display the test performances of the models on the classification and regression tasks respectively. All regression tasks are evaluated using the $R^2$ score, while the classification tasks are evaluated using the AUROC score. We report the mean values of the metrics over 5 runs for each task, as well as the standard deviation.

We display in Figure 12 the evolution of the average rank of the embedders when separating the tasks based on the amount of samples, and the class imbalance (for classification tasks). Our student appears robust in both setups, even though as the class imbalance becomes more important, or as the amount of samples in the task decreases, the difference between the top-performing embedders becomes less significant.

Table 6: AUROC of each model on the ADMET and HTS downstream classification tasks. The best embedder for each task is highlighted in bold and underlined, and the second best is highlighted in bold.

| Model | Metabolism CYP3A4 | Metabolism CYP3A4 (s) | Metabolism CYP2D6 | Metabolism CYP2D6 (s) | Metabolism CYP2C9 | Metabolism CYP2C9 (s) | Metabolism CYP2C19 | Metabolism CYP1A2 | HTS HIV | Distribution BBB | avg |
|---|---|---|---|---|---|---|---|---|---|---|---|
| InfoGraph | 0.817±0.003 | 0.567±0.042 | 0.813±0.013 | 0.674±0.031 | 0.840±0.007 | 0.624±0.085 | 0.832±0.004 | 0.878±0.003 | 0.769±0.018 | 0.843±0.022 | 0.768±0.097 |
| ChemGPT-1.2B | 0.844±0.006 | 0.608±0.026 | 0.816±0.020 | 0.670±0.014 | 0.852±0.008 | 0.638±0.034 | 0.835±0.004 | 0.886±0.008 | 0.760±0.014 | 0.853±0.012 | 0.779±0.094 |
| FRAD QM9[t] | 0.855±0.003 | 0.607±0.060 | 0.815±0.018 | 0.711±0.025 | 0.860±0.006 | 0.609±0.049 | 0.845±0.010 | 0.906±0.004 | 0.779±0.005 | 0.869±0.013 | 0.785±0.111 |
| ChemBertMLM-10M | 0.846±0.005 | 0.606±0.029 | 0.813±0.014 | 0.733±0.017 | 0.851±0.009 | 0.643±0.020 | 0.843±0.007 | 0.886±0.007 | 0.733±0.012 | 0.868±0.009 | 0.785±0.089 |
| GROVER | 0.827±0.007 | 0.599±0.028 | 0.824±0.013 | 0.750±0.040 | 0.853±0.005 | 0.614±0.042 | 0.843±0.008 | 0.880±0.006 | 0.760±0.014 | 0.869±0.016 | 0.787±0.096 |
| GraphCL[t] | 0.846±0.008 | 0.607±0.044 | 0.828±0.009 | 0.723±0.030 | 0.862±0.006 | 0.645±0.032 | 0.858±0.007 | 0.898±0.004 | 0.765±0.019 | 0.865±0.011 | 0.792±0.093 |
| GraphLog[t] | 0.847±0.009 | 0.624±0.049 | 0.832±0.017 | **0.768**±0.046 | 0.860±0.009 | 0.616±0.034 | 0.855±0.006 | 0.886±0.006 | 0.748±0.008 | 0.867±0.017 | 0.790±0.096 |
| GraphMVP[t] | 0.847±0.011 | 0.619±0.034 | 0.830±0.016 | 0.747±0.033 | 0.870±0.005 | 0.622±0.082 | 0.865±0.006 | 0.899±0.004 | 0.771±0.016 | 0.874±0.016 | 0.800±0.097 |
| MolR_gat | 0.873±0.005 | 0.598±0.045 | 0.841±0.017 | 0.717±0.017 | 0.869±0.004 | 0.615±0.037 | 0.859±0.010 | 0.909±0.004 | 0.797±0.012 | 0.867±0.025 | 0.808±0.098 |
| ThreeDInfomax[t] | 0.858±0.009 | 0.604±0.015 | 0.842±0.010 | 0.747±0.039 | 0.865±0.009 | 0.571±0.085 | 0.868±0.008 | 0.917±0.005 | 0.762±0.010 | **0.885**±0.019 | 0.815±0.100 |
| ChemBertMTR-77M[t] | 0.883±0.008 | **0.645**±0.051 | 0.845±0.015 | 0.734±0.024 | 0.877±0.007 | 0.600±0.114 | 0.873±0.005 | 0.919±0.007 | 0.797±0.014 | **0.894**±0.026 | 0.816±0.096 |
| L2 | 0.871±0.008 | 0.641±0.063 | 0.849±0.012 | 0.742±0.034 | 0.887±0.008 | 0.663±0.049 | 0.864±0.010 | 0.917±0.003 | 0.797±0.005 | 0.878±0.022 | 0.806±0.094 |
| Cosine | **0.892**±0.005 | 0.637±0.025 | 0.855±0.015 | 0.758±0.045 | 0.887±0.008 | 0.673±0.070 | 0.879±0.005 | 0.925±0.003 | 0.786±0.014 | 0.881±0.014 | 0.816±0.096 |
| student-250k | **0.893**±0.009 | **0.646**±0.052 | **0.865**±0.017 | **0.767**±0.040 | **0.893**±0.005 | **0.699**±0.080 | **0.881**±0.005 | **0.931**±0.005 | 0.796±0.013 | 0.875±0.020 | **0.823**±0.095 |
| student-2M | **0.892**±0.004 | 0.639±0.054 | **0.868**±0.010 | 0.766±0.058 | **0.893**±0.002 | **0.697**±0.093 | **0.882**±0.006 | **0.933**±0.006 | **0.800**±0.014 | 0.882±0.020 | **0.825**±0.096 |

| Model | Tox hERG (k) | Tox hERG | Tox Tox21 | Tox Skin R | Tox DILI | Tox ClinTox | Tox Carcinogens | Tox AMES | Absorption Pgp | Absorption PAMPA | Absorption HIA | Absorption Bioavailability |
|---|---|---|---|---|---|---|---|---|---|---|---|---|
| InfoGraph | 0.849±0.009 | 0.778±0.027 | 0.770±0.058 | 0.714±0.030 | 0.837±0.056 | 0.621±0.086 | 0.728±0.042 | 0.853±0.009 | 0.896±0.022 | 0.685±0.031 | 0.872±0.085 | 0.631±0.015 |
| ChemGPT-1.2B | 0.867±0.007 | 0.789±0.049 | 0.762±0.066 | 0.721±0.077 | 0.857±0.031 | 0.641±0.022 | 0.785±0.017 | 0.843±0.012 | 0.926±0.027 | 0.665±0.050 | 0.859±0.055 | 0.668±0.046 |
| FRAD QM9[t] | 0.873±0.004 | 0.817±0.032 | 0.797±0.060 | 0.713±0.043 | 0.843±0.004 | 0.553±0.054 | 0.772±0.063 | 0.871±0.009 | 0.914±0.024 | 0.699±0.043 | 0.945±0.034 | 0.626±0.022 |
| ChemBertMLM-10M | 0.867±0.007 | 0.779±0.017 | 0.789±0.055 | 0.747±0.087 | 0.791±0.060 | 0.648±0.082 | 0.776±0.063 | 0.858±0.013 | 0.911±0.029 | 0.715±0.056 | 0.892±0.082 | 0.664±0.069 |
| GROVER | 0.856±0.004 | 0.774±0.034 | 0.780±0.039 | 0.749±0.081 | 0.844±0.036 | 0.637±0.053 | 0.779±0.084 | 0.867±0.012 | 0.918±0.012 | 0.703±0.027 | 0.931±0.038 | 0.663±0.038 |
| GraphCL[t] | 0.864±0.005 | 0.799±0.038 | 0.787±0.058 | 0.770±0.038 | 0.827±0.035 | 0.639±0.078 | **0.847**±0.064 | 0.869±0.016 | 0.920±0.030 | 0.709±0.034 | 0.863±0.052 | 0.643±0.027 |
| GraphLog[t] | 0.849±0.006 | 0.797±0.049 | 0.801±0.052 | 0.751±0.101 | 0.853±0.035 | 0.696±0.081 | 0.793±0.076 | 0.869±0.006 | 0.920±0.026 | 0.637±0.024 | 0.897±0.035 | 0.622±0.071 |
| GraphMVP[t] | 0.872±0.005 | 0.823±0.035 | 0.793±0.035 | 0.750±0.087 | 0.867±0.049 | 0.624±0.046 | 0.779±0.095 | 0.874±0.011 | 0.918±0.038 | 0.718±0.009 | 0.944±0.025 | **0.694**±0.055 |
| MolR_gat | 0.881±0.011 | **0.844**±0.022 | 0.800±0.061 | 0.748±0.047 | 0.858±0.042 | 0.810±0.048 | 0.760±0.087 | 0.871±0.014 | 0.928±0.028 | 0.705±0.061 | 0.957±0.020 | 0.672±0.049 |
| ThreeDInfomax[t] | 0.874±0.010 | 0.829±0.035 | 0.804±0.039 | **0.833**±0.041 | 0.842±0.037 | **0.837**±0.043 | 0.791±0.074 | 0.872±0.018 | 0.929±0.030 | 0.745±0.026 | **0.986**±0.014 | 0.670±0.033 |
| ChemBertMTR-77M[t] | 0.897±0.009 | 0.832±0.034 | **0.818**±0.063 | 0.758±0.069 | 0.858±0.037 | 0.734±0.068 | 0.776±0.033 | 0.881±0.005 | **0.936**±0.030 | **0.763**±0.026 | 0.960±0.034 | **0.683**±0.027 |
| L2 | 0.895±0.007 | 0.824±0.018 | 0.807±0.061 | 0.770±0.039 | 0.856±0.033 | 0.654±0.094 | 0.783±0.019 | 0.871±0.010 | 0.914±0.030 | 0.735±0.027 | 0.914±0.040 | 0.626±0.076 |
| Cosine | 0.908±0.006 | 0.830±0.038 | 0.814±0.056 | 0.780±0.033 | **0.879**±0.030 | 0.650±0.092 | 0.822±0.084 | 0.884±0.008 | 0.926±0.021 | **0.755**±0.024 | 0.908±0.062 | 0.629±0.043 |
| student-250k | **0.912**±0.006 | 0.828±0.050 | 0.817±0.062 | 0.812±0.040 | 0.858±0.036 | 0.633±0.082 | 0.810±0.079 | **0.888**±0.009 | 0.926±0.025 | 0.742±0.025 | **0.969**±0.018 | 0.671±0.043 |
| student-2M | **0.911**±0.005 | **0.834**±0.019 | **0.819**±0.054 | 0.801±0.026 | 0.856±0.045 | 0.662±0.072 | **0.839**±0.095 | **0.891**±0.014 | **0.936**±0.024 | 0.730±0.024 | 0.959±0.026 | 0.653±0.055 |

Table 7: $R^2$ score of each model on the ADMET downstream regression tasks. The best embedder for each task is highlighted in bold and underlined, and the second best is highlighted in bold.

| | avg | Absorption Caco2 | Absorption FreeSolv | Absorption Lipophilicity | Absorption Solubility |
|---|---|---|---|---|---|
| InfoGraph | $0.275\pm_{0.284}$ | $0.491\pm_{0.031}$ | $0.639\pm_{0.058}$ | $0.341\pm_{0.035}$ | $0.700\pm_{0.007}$ |
| ChemBertMLM-10M | $0.264\pm_{0.364}$ | $0.543\pm_{0.076}$ | $0.776\pm_{0.038}$ | $0.363\pm_{0.063}$ | $0.774\pm_{0.007}$ |
| FRAD QM9$^{(t)}$ | $0.332\pm_{0.284}$ | $0.564\pm_{0.051}$ | $0.686\pm_{0.082}$ | $0.483\pm_{0.029}$ | $0.758\pm_{0.011}$ |
| ChemGPT-1.2B | $0.340\pm_{0.329}$ | $0.567\pm_{0.079}$ | $0.831\pm_{0.048}$ | $0.487\pm_{0.020}$ | $0.798\pm_{0.009}$ |
| GROVER | $0.350\pm_{0.274}$ | $0.575\pm_{0.058}$ | $0.708\pm_{0.024}$ | $0.470\pm_{0.043}$ | $0.733\pm_{0.027}$ |
| GraphLog$^{(t)}$ | $0.350\pm_{0.311}$ | $0.545\pm_{0.055}$ | $0.811\pm_{0.017}$ | $0.486\pm_{0.037}$ | $0.765\pm_{0.010}$ |
| GraphCL$^{(t)}$ | $0.355\pm_{0.292}$ | $0.559\pm_{0.051}$ | $0.764\pm_{0.038}$ | $0.467\pm_{0.067}$ | $0.745\pm_{0.021}$ |
| GraphMVP$^{(t)}$ | $0.397\pm_{0.320}$ | $0.592\pm_{0.064}$ | $0.861\pm_{0.036}$ | $0.590\pm_{0.064}$ | $0.791\pm_{0.009}$ |
| MolR gat | $0.394\pm_{0.307}$ | $0.651\pm_{0.089}$ | $0.804\pm_{0.075}$ | $0.518\pm_{0.037}$ | $0.822\pm_{0.010}$ |
| ThreeDInfomax$^{(t)}$ | $0.425\pm_{0.322}$ | $0.700\pm_{0.038}$ | $0.852\pm_{0.055}$ | $0.624\pm_{0.031}$ | $\mathbf{0.848}\pm_{\mathbf{0.004}}$ |
| ChemBertMTR-77M$^{(t)}$ | $0.459\pm_{0.308}$ | $\underline{\mathbf{0.725}}\pm_{\mathbf{0.027}}$ | $0.874\pm_{0.037}$ | $0.670\pm_{0.025}$ | $\mathbf{0.839}\pm_{\mathbf{0.007}}$ |
| L2 | $0.420\pm_{0.299}$ | $0.642\pm_{0.060}$ | $0.851\pm_{0.063}$ | $0.605\pm_{0.021}$ | $0.792\pm_{0.018}$ |
| Cosine | $0.460\pm_{0.311}$ | $0.699\pm_{0.056}$ | $\mathbf{0.893}\pm_{\mathbf{0.034}}$ | $0.721\pm_{0.028}$ | $0.815\pm_{0.009}$ |
| student-250k | $\underline{\mathbf{0.482}}\pm_{\mathbf{0.298}}$ | $\mathbf{0.712}\pm_{\mathbf{0.040}}$ | $\underline{\mathbf{0.900}}\pm_{\mathbf{0.035}}$ | $\underline{\mathbf{0.742}}\pm_{\mathbf{0.019}}$ | $0.823\pm_{0.007}$ |
| student-2M | $\mathbf{0.476}\pm_{\mathbf{0.301}}$ | $0.687\pm_{0.045}$ | $0.878\pm_{0.036}$ | $\mathbf{0.739}\pm_{\mathbf{0.021}}$ | $0.822\pm_{0.005}$ |

| | Distribution PPBR | Distribution VDss | Excretion Clearance (H) | Excretion Clearance (M) | Excretion Half Life | Tox LD50 |
|---|---|---|---|---|---|---|
| InfoGraph | $0.093\pm_{0.073}$ | $0.018\pm_{0.190}$ | $-0.048\pm_{0.133}$ | $0.070\pm_{0.046}$ | $-0.011\pm_{0.161}$ | $0.458\pm_{0.039}$ |
| ChemBertMLM-10M | $0.112\pm_{0.035}$ | $0.066\pm_{0.091}$ | $-0.185\pm_{0.122}$ | $0.040\pm_{0.178}$ | $-0.240\pm_{0.279}$ | $0.390\pm_{0.044}$ |
| FRAD QM9$^{(t)}$ | $0.180\pm_{0.031}$ | $-0.004\pm_{0.050}$ | $0.006\pm_{0.095}$ | $0.124\pm_{0.059}$ | $0.104\pm_{0.129}$ | $0.415\pm_{0.039}$ |
| ChemGPT-1.2B | $0.175\pm_{0.036}$ | $0.046\pm_{0.173}$ | $-0.018\pm_{0.071}$ | $0.117\pm_{0.099}$ | $-0.047\pm_{0.182}$ | $0.442\pm_{0.043}$ |
| GROVER | $0.185\pm_{0.056}$ | $\mathbf{0.186}\pm_{\mathbf{0.079}}$ | $-0.034\pm_{0.095}$ | $0.197\pm_{0.082}$ | $0.035\pm_{0.161}$ | $0.447\pm_{0.058}$ |
| GraphLog$^{(t)}$ | $0.240\pm_{0.082}$ | $\underline{\mathbf{0.202}}\pm_{\mathbf{0.111}}$ | $-0.094\pm_{0.053}$ | $0.068\pm_{0.120}$ | $0.018\pm_{0.192}$ | $0.457\pm_{0.054}$ |
| GraphCL$^{(t)}$ | $0.237\pm_{0.048}$ | $0.158\pm_{0.075}$ | $-0.022\pm_{0.127}$ | $0.123\pm_{0.108}$ | $0.007\pm_{0.165}$ | $0.508\pm_{0.026}$ |
| GraphMVP$^{(t)}$ | $0.327\pm_{0.036}$ | $0.168\pm_{0.081}$ | $-0.009\pm_{0.135}$ | $0.144\pm_{0.071}$ | $-0.017\pm_{0.226}$ | $0.527\pm_{0.042}$ |
| MolR gat | $0.284\pm_{0.093}$ | $0.155\pm_{0.180}$ | $-0.024\pm_{0.091}$ | $0.174\pm_{0.050}$ | $0.059\pm_{0.232}$ | $0.496\pm_{0.040}$ |
| ThreeDInfomax$^{(t)}$ | $0.314\pm_{0.053}$ | $0.152\pm_{0.061}$ | $0.071\pm_{0.049}$ | $0.195\pm_{0.114}$ | $-0.004\pm_{0.264}$ | $0.500\pm_{0.040}$ |
| ChemBertMTR-77M$^{(t)}$ | $\mathbf{0.393}\pm_{\mathbf{0.055}}$ | $0.138\pm_{0.127}$ | $0.011\pm_{0.048}$ | $0.250\pm_{0.078}$ | $\mathbf{0.196}\pm_{\mathbf{0.190}}$ | $0.491\pm_{0.031}$ |
| L2 | $0.362\pm_{0.077}$ | $0.135\pm_{0.097}$ | $0.034\pm_{0.097}$ | $0.244\pm_{0.062}$ | $0.060\pm_{0.116}$ | $0.470\pm_{0.030}$ |
| Cosine | $0.382\pm_{0.032}$ | $0.108\pm_{0.084}$ | $\mathbf{0.079}\pm_{\mathbf{0.102}}$ | $0.275\pm_{0.054}$ | $0.111\pm_{0.158}$ | $0.515\pm_{0.039}$ |
| student-250k | $\underline{\mathbf{0.390}}\pm_{\mathbf{0.042}}$ | $0.125\pm_{0.111}$ | $\underline{\mathbf{0.113}}\pm_{\mathbf{0.070}}$ | $0.283\pm_{0.076}$ | $\underline{\mathbf{0.207}}\pm_{\mathbf{0.101}}$ | $\mathbf{0.529}\pm_{\mathbf{0.039}}$ |
| student-2M | $0.389\pm_{0.050}$ | $0.138\pm_{0.115}$ | $0.069\pm_{0.060}$ | $\underline{\mathbf{0.348}}\pm_{\mathbf{0.062}}$ | $0.144\pm_{0.205}$ | $\underline{\mathbf{0.543}}\pm_{\mathbf{0.041}}$ |

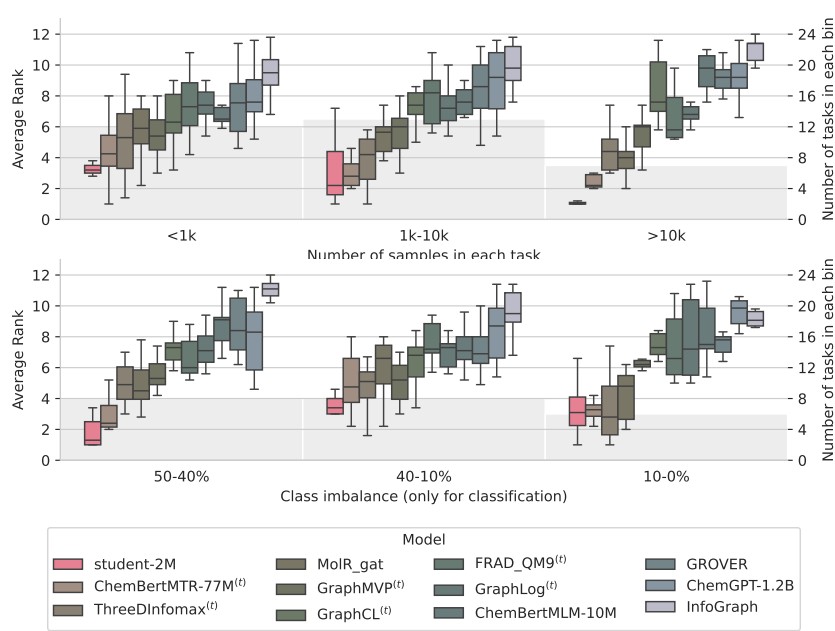

Figure 12: Average ranking of our models when grouping tasks based on the number of samples in the task and the class imbalance (for classification tasks).

# C NATURAL LANGUAGE PROCESSING

## C.1 TRAINING SET AND HYPERPARAMETERS

### C.1.1 TRAINING SET

**Dataset sources.** We ran experiments with two training sets a home-made dataset combining different training sets of different embedders and the GISTEmbed dataset. We provide the statistics of our dataset in Tab. 8 and the GISTEmbed dataset is described in Solatorio (2024).

**Dataset construction.** Most embedding datasets consists of positive and negative samples, questions and answers, or sentences and their labels. We flattened the datasets to have only one column of sentences and deduplicated the dataset. For the MEDI dataset for example, given query, positive and negative samples we build a dataset with three times the number of entries, one for each sentence. We then deduplicated the dataset to remove any duplicate entries.

Table 8: Number of samples in each dataset

| URL | Number of samples |
| --- | --- |
| https://huggingface.co/datasets/embedding-data/SPECTER | 190872 |
| https://huggingface.co/datasets/embedding-data/Amazon-QA | 3264474 |
| https://huggingface.co/datasets/embedding-data/simple-wiki | 203755 |
| https://huggingface.co/datasets/embedding-data/QQP_triplets | 328188 |
| https://huggingface.co/datasets/embedding-data/sentence-compression | 356409 |
| https://huggingface.co/datasets/embedding-data/altlex | 223901 |
| https://huggingface.co/datasets/fancyzhx/ag_news | 120000 |
| https://huggingface.co/datasets/stanfordnlp/sst2 | 67349 |
| https://huggingface.co/datasets/dair-ai/emotion | 416809 |
| https://huggingface.co/datasets/stanfordnlp/snli | 1100304 |
| https://huggingface.co/datasets/cardiffnlp/tweet_eval | 45000 |
| https://huggingface.co/datasets/stanfordnlp/imdb | 25000 |
| | 6342061 |

Table 9: Performance of the 4 teachers we used and of the base students. Experiments with single teacher distillation were performed with the stronger teacher SFR-Embedding-2_R.

| | | Size | Amazon Counterfactual | Amazon Polarity | Amazon Reviews | Banking77 | Emotion | Imdb | MTOPDomain | MTOPIntent | Massive Intent | Massive Scenario | Toxic Conversations | Tweet Sentiment Extraction | Avg. |
|---|---|---|---|---|---|---|---|---|---|---|---|---|---|---|---|
| Teacher | SFR-Embedding-2_R | 7111.0 | 92.7 | 97.3 | 61.0 | 90.0 | 93.4 | 96.8 | 98.6 | 91.3 | 86.0 | 90.6 | 91.1 | 79.7 | 89.0 |
| | stella_en_400M_v5 | 435.0 | 92.4 | 97.2 | 59.5 | 89.3 | 78.8 | 96.5 | 98.8 | 92.3 | 85.2 | 89.6 | 86.9 | 73.6 | 86.7 |
| | UAE-Large-V1 | 335.0 | 75.5 | 92.8 | 48.3 | 87.7 | 51.8 | 92.8 | 94.0 | 76.9 | 76.5 | 79.8 | 71.1 | 59.8 | 75.6 |
| | sf_model_e5 | 335.0 | 70.8 | 91.8 | 48.9 | 84.6 | 54.9 | 93.1 | 93.6 | 66.0 | 73.5 | 77.4 | 71.2 | 61.5 | 74.0 |
| Student (Base) | snowflake-arctic-embed-m | 109.0 | 76.8 | 82.8 | 38.9 | 80.3 | 46.5 | 74.1 | 92.7 | 65.2 | 66.9 | 72.8 | 64.9 | 56.7 | 68.2 |
| | snowflake-arctic-embed-s | 33.0 | 71.2 | 78.8 | 38.3 | 79.1 | 45.8 | 69.5 | 90.9 | 58.6 | 64.8 | 70.0 | 62.0 | 58.9 | 65.7 |
| | snowflake-arctic-embed-xs | 23.0 | 65.1 | 70.0 | 35.3 | 76.4 | 41.8 | 62.8 | 90.8 | 58.0 | 63.5 | 71.0 | 64.3 | 56.2 | 62.9 |

### C.1.2 TEACHERS AND BASED STUDENTS PERFORMANCE

**Teachers.** We selected 4 teachers from the MTEB benchmark Muennighoff et al. (2023) as teachers for our distillation method. We provide the list of the teachers and their performance in Tab. 9. The 4 teachers of widely different sizes (335M, 435M and 7B) have display strong but different performances on the MTEB benchmark.

### C.1.3 SINGLE TEACHER DISTILLATION

**Single teacher vs. Multi-Teachers.** Since some teachers yield strong performance on their own, distilling only from the strongest could yield similar results as the multi-teacher setting involving weaker teachers. We applied our method in a single-teacher setting using the strongest teacher by far (SF-Embeddings-R_2) as a teacher and compared the results to the multi-teacher setting. Consistently with results in computer vision and molecular representations, we found that adding weaker teachers did improve our results (Figure 13), supporting our hypothesis that enforcing reconstruction capabilities for a diversity of models indeed leads to more informative representations.

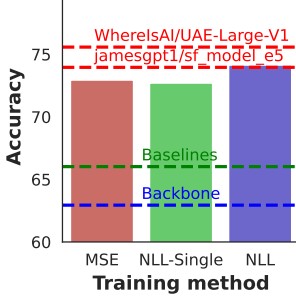

Figure 13: Comparison of distilled small model with the performance of the initial backbone, baselines in the MTEB, with our teachers' performance.

### C.1.4 HYPERPARAMETERS

**Training hyperparameters.** We trained our models using the Adam optimizer with a learning rate of $5.10^{-5}$ and an effective batch size of $2048$ for all our models using different number of accumulation steps and batch size depending on the models' sizes. We did not use any learning rate scheduler.

## C.2 DETAILED EVALUATION RESULTS

We ran different parts of the MTEB benchmarks and report the overall results for all our models in this section.

### C.2.1 EVALUATION ON CLASSIFICATION TASKS

**Small models' performance.** In Tab. 10 and Tab. 11, we provide the classification accuracy of our distilled models on the MTEB classification benchmark for our smaller models xs (22M) and s (33M). Our smallest model significantly improves SOTA performance for models of its size by increasing the average score of 2 points compared to the previous best model.

Table 10: Performance of our distilled models compared of models of similar sizes 16M to 30M parameters from the MTEB Benchmark on classification tasks.

| | Task Model | Size | Amazon Counterfactual | Amazon Polarity | Amazon Reviews | Banking77 | Emotion | Imdb | MTOPDomain | MTOPIntent | Massive Intent | Massive Scenario | Toxic Conversations | Tweet Sentiment Extraction | Avg. |
|---|---|---|---|---|---|---|---|---|---|---|---|---|---|---|---|
| | GIST | 23M | 72.9 | **87.2** | 42.6 | 84.2 | 52.1 | 78.5 | 94.8 | 77.7 | 73.2 | 76.7 | **72.9** | 59.9 | 72.7 |
| | Bulbasaur | 17M | 71.9 | 78.8 | 39.3 | 80.6 | 44.8 | 71.5 | 90.8 | 68.7 | 68.8 | 73.8 | 66.3 | 59.5 | 67.9 |
| | Ivysaur | 23M | 72.1 | 86.7 | **42.7** | 81.9 | 45.4 | 80.8 | 92.1 | 71.9 | 70.3 | 74.9 | 65.5 | 58.7 | 70.2 |
| | Squirtle | 16M | 69.6 | 82.1 | 41.9 | 67.1 | 45.8 | 75.0 | 87.3 | 54.7 | 61.5 | 67.0 | 64.5 | 61.8 | 64.9 |
| | Venusaur | 16M | 73.2 | 80.0 | 39.7 | 78.0 | 44.4 | 73.0 | 89.9 | 71.0 | 67.8 | 72.4 | 64.4 | 59.7 | 67.8 |
| | Wartortle | 17M | 70.4 | 82.0 | 42.4 | 71.1 | 46.8 | 74.6 | 88.2 | 54.9 | 62.3 | 68.2 | 65.2 | 62.5 | 65.7 |
| | gte-micro | 17M | 68.8 | 77.1 | 40.9 | 69.6 | 46.2 | 62.2 | 86.7 | 49.7 | 59.0 | 66.6 | 66.1 | 60.8 | 62.8 |
| MTEB | gte-micro-v2 | 17M | 71.4 | 77.7 | 39.0 | 80.4 | 44.5 | 70.6 | 90.5 | 67.5 | 68.5 | 73.5 | 66.7 | 59.3 | 67.5 |
| | gte-micro-v4 | 19M | 71.8 | 80.0 | 39.8 | 80.9 | 44.9 | 72.0 | 90.9 | 68.5 | 69.1 | 74.2 | 66.0 | 59.4 | 68.1 |
| | snowflake-arctic-embed-xs | 23M | 65.1 | 70.0 | 35.3 | 76.4 | 41.8 | 62.8 | 90.8 | 58.0 | 63.5 | 71.0 | 64.3 | 56.2 | 62.9 |
| | bge-micro | 17M | 66.3 | 75.4 | 35.8 | 80.6 | 42.5 | 70.7 | 90.2 | 68.0 | 67.8 | 73.0 | 69.2 | 56.7 | 66.3 |
| | bge-micro-v2 | 17M | 67.8 | 79.8 | 37.5 | 81.2 | 44.5 | 76.5 | 90.7 | 68.3 | 68.6 | 73.9 | 70.2 | 57.6 | 68.0 |
| | gte-tiny | 23M | 71.8 | 86.6 | 42.6 | 81.7 | 44.7 | 80.5 | 91.8 | 69.9 | 70.1 | 74.9 | 71.0 | 58.6 | 70.3 |
| | slx-v0.1 | 23M | 61.5 | 64.3 | 30.3 | 80.0 | 40.5 | 61.8 | 92.0 | 63.3 | 67.9 | 73.9 | 62.1 | 54.0 | 62.6 |
| | multi-qa-MiniLM-L6-cos-v1 | 23M | 61.8 | 62.4 | 29.6 | 78.6 | 39.6 | 61.2 | 90.0 | 59.6 | 66.8 | 73.8 | 65.1 | 51.6 | 61.7 |
| | all-MiniLM-L6-v2 | 23M | 63.6 | 64.3 | 34.0 | 80.0 | 40.8 | 61.8 | 91.7 | 61.5 | 66.9 | 73.8 | 62.1 | 54.0 | 62.6 |
| MSE | Student-xs | 23M | 71.6 | 86.2 | 42.3 | 83.6 | 57.5 | **83.5** | 94.5 | 75.4 | 74.3 | **80.4** | 66.3 | 59.3 | 72.9 |
| NLL | Student-xs | 23M | **76.5** | 84.9 | 42.4 | **85.8** | **58.0** | 81.1 | **95.2** | **79.9** | **75.8** | 80.4 | 68.1 | 60.1 | **74.0** |

Table 11: Performance of our distilled models compared of models of similar sizes 30M to 50M parameters from the MTEB Benchmark on classification tasks.

| | Task Model | Size | Amazon Counterfactual | Amazon Polarity | Amazon Reviews | Banking77 | Emotion | Imdb | MTOPDomain | MTOPIntent | Massive Intent | Massive Scenario | Toxic Conversations | Tweet Sentiment Extraction | Avg. |
|---|---|---|---|---|---|---|---|---|---|---|---|---|---|---|---|
| | bge-small-en-v1.5 | 33M | 73.8 | 92.8 | 47.0 | 85.7 | 47.8 | **90.6** | 93.4 | 74.8 | 74.8 | 78.7 | 69.9 | 60.5 | 74.1 |
| | GIST | 33M | 75.3 | 93.2 | 49.7 | 86.7 | 55.9 | 89.5 | 95.5 | 79.1 | 75.5 | 79.2 | 72.8 | 61.0 | **76.1** |
| | snowflake-arctic-embed-s | 33M | 71.2 | 78.8 | 38.3 | 79.1 | 45.8 | 69.5 | 90.9 | 58.6 | 64.8 | 70.0 | 62.0 | 58.9 | 65.7 |
| | bge-small-4096 | 35M | 68.8 | 81.3 | 38.6 | 80.0 | 40.1 | 80.1 | 90.4 | 66.5 | 67.6 | 73.5 | 69.3 | 57.6 | 67.8 |
| | NoInstruct-small-Embedding-v0 | 33M | 75.8 | **93.3** | **50.0** | 86.4 | 55.1 | 90.2 | 95.3 | 79.6 | 76.0 | 79.3 | 69.4 | 61.3 | 76.0 |
| MTEB | LASER | 43M | 76.8 | 61.0 | 28.7 | 57.8 | 24.8 | 57.6 | 75.4 | 49.5 | 47.9 | 55.9 | 54.0 | 48.7 | 53.2 |
| | e5-small | 33M | 76.2 | 87.5 | 42.6 | 81.9 | 46.9 | 75.5 | 92.0 | 73.2 | 72.2 | 75.8 | 72.8 | 63.3 | 71.7 |
| | e5-small-v2 | 33M | 77.6 | 91.3 | 45.9 | 81.6 | 47.1 | 86.0 | 92.7 | 72.6 | 71.6 | 76.4 | 71.1 | 61.5 | 72.9 |
| | jina-embedding-s-en-v1 | 35M | 64.8 | 64.3 | 30.6 | 74.6 | 36.1 | 58.7 | 88.8 | 58.6 | 64.7 | 71.8 | 59.4 | 54.3 | 60.6 |
| | jina-embeddings-v2-small-en | 33M | 71.4 | 82.9 | 40.9 | 78.2 | 44.0 | 73.6 | 94.0 | 72.5 | 67.6 | 69.8 | 71.5 | 59.4 | 68.8 |
| | all-MiniLM-L12-v2 | 33M | 65.3 | 63.0 | 30.8 | 80.4 | 41.2 | 59.8 | 91.9 | 62.8 | 67.2 | 74.6 | 67.5 | 54.2 | 63.2 |
| | gte-small | 33M | 73.2 | 91.8 | 48.0 | 84.1 | 46.6 | 86.8 | 93.0 | 69.7 | 70.3 | 75.6 | 70.3 | 58.2 | 72.3 |
| MSE | Student-s | 33M | 72.6 | 90.3 | 44.3 | 84.2 | 56.5 | 88.8 | 94.9 | 77.2 | 75.4 | **81.2** | 64.9 | 60.4 | 74.2 |
| NLL | Student-s | 33M | 77.3 | 89.2 | 43.8 | **86.7** | **58.0** | 88.3 | **95.5** | **81.9** | **76.7** | 80.7 | 66.1 | 60.6 | 75.4 |

Table 12: Performance of our distilled models compared of models of similar sizes 100M to 120M parameters from the MTEB Benchmark on classification tasks.

| | Task
Model | Size | Amazon Counterfactual | Amazon Polarity | Amazon Reviews | Banking77 | Emotion | Imdb | MTOPDomain | MTOPIntent | Massive Intent | Massive Scenario | Toxic Conversations | Tweet Sentiment Extraction | Avg. |
|---|---|---|---|---|---|---|---|---|---|---|---|---|---|---|---|
| MTEB | bge-base-en-v1.5 | 109M | 76.2 | 93.4 | 48.9 | 87.0 | 51.9 | **90.8** | 94.2 | 76.9 | 76.2 | 80.2 | 71.6 | 59.4 | 75.5 |
| | GIST | 109M | 76.0 | **93.5** | **50.5** | 87.3 | 54.7 | 89.7 | 95.3 | 78.1 | 76.0 | 79.6 | 72.4 | 59.3 | 76.0 |
| | bilingual-embedding-small | 118M | 74.3 | 82.2 | 40.2 | 80.3 | 40.8 | 73.7 | 89.7 | 66.5 | 68.9 | 74.5 | 62.5 | 59.6 | 67.8 |
| | multilingual-e5-small | 118M | 73.8 | 88.7 | 44.7 | 79.4 | 42.5 | 80.8 | 91.1 | 71.1 | 70.3 | 74.5 | 69.4 | 62.6 | 70.7 |
| | snowflake-arctic-embed-m | 109M | 76.8 | 82.8 | 38.9 | 80.3 | 46.5 | 74.1 | 92.7 | 65.2 | 66.9 | 72.8 | 64.9 | 56.7 | 68.2 |
| | snowflake-arctic-embed-m-v1.5 | 109M | 68.3 | 90.3 | 46.3 | 80.0 | 43.7 | 84.4 | 91.4 | 60.6 | 66.7 | 73.1 | 66.8 | 53.9 | 68.8 |
| | ml-nlp-elser.html | 110M | 74.2 | 61.9 | 32.1 | 82.0 | 46.6 | 65.0 | 93.2 | 71.1 | 68.5 | 75.0 | 68.2 | 53.6 | 65.9 |
| | e5-base-4k | 112M | 77.8 | 92.8 | 46.7 | 83.5 | 47.0 | 86.2 | 93.7 | 75.3 | 73.0 | 77.7 | 72.1 | 60.4 | 73.8 |
| | instructor-base | 110M | **86.2** | 88.4 | 44.6 | 77.0 | 51.8 | 81.2 | 93.7 | 70.3 | 67.5 | 72.6 | 71.8 | **63.3** | 72.4 |
| | bert-base-uncased | 110M | 74.2 | 71.3 | 33.6 | 63.4 | 35.3 | 65.3 | 82.6 | 68.1 | 59.9 | 64.3 | 70.0 | 51.8 | 61.7 |
| | e5-base | 109M | 79.7 | 88.0 | 42.6 | 83.3 | 49.4 | 76.0 | 93.2 | 74.8 | 72.2 | 76.8 | 74.1 | 61.4 | 72.6 |
| | e5-base-v2 | 110M | 77.8 | 92.8 | 46.7 | 83.5 | 47.0 | 86.2 | 93.7 | 75.3 | 73.0 | 77.7 | 72.1 | 60.4 | 73.8 |
| | jina-embedding-b-en-v1 | 110M | 66.7 | 67.6 | 31.2 | 84.1 | 44.7 | 63.9 | 91.5 | 72.8 | 71.1 | 76.2 | 66.2 | 56.9 | 66.1 |
| | contriever-base-msmarco | 110M | 72.2 | 68.6 | 37.4 | 80.0 | 44.8 | 67.0 | 93.2 | 69.3 | 67.8 | 76.0 | 67.8 | 56.1 | 66.7 |
| | sup-simcse-bert-base-uncased | 110M | 75.8 | 82.5 | 39.6 | 75.8 | 44.8 | 73.5 | 84.3 | 63.1 | 66.0 | 70.8 | 72.0 | 59.7 | 67.3 |
| | unsup-simcse-bert-base-uncased | 110M | 67.1 | 74.5 | 33.9 | 73.5 | 42.2 | 69.6 | 81.7 | 59.2 | 59.8 | 66.2 | 68.8 | 53.4 | 62.5 |
| | all-mpnet-base-v2 | 110M | 65.0 | 67.1 | 31.4 | 81.7 | 42.2 | 71.2 | 91.9 | 68.3 | 69.8 | 75.7 | 61.0 | 55.0 | 65.0 |
| | allenai-specter | 110M | 58.7 | 57.8 | 26.3 | 66.7 | 24.8 | 56.4 | 74.5 | 50.0 | 51.7 | 58.6 | 57.4 | 45.5 | 52.4 |
| | gtr-t5-base | 110M | 69.3 | 67.8 | 38.5 | 79.3 | 42.2 | 66.0 | 92.4 | 62.4 | 67.0 | 75.4 | 66.6 | 56.0 | 65.3 |
| | msmarco-bert-co-condensor | 110M | 64.1 | 66.9 | 34.9 | 82.3 | 41.9 | 60.2 | 91.3 | 71.1 | 70.4 | 73.7 | 64.0 | 55.7 | 64.7 |
| | paraphrase-multilingual-MiniLM-L12-v2 | 118M | 71.5 | 69.2 | 35.1 | 79.8 | 42.3 | 60.5 | 87.0 | 65.5 | 66.9 | 71.5 | 60.1 | 56.1 | 63.8 |
| | sentence-t5-base | 110M | 75.8 | 85.1 | 44.9 | 76.5 | 51.4 | 77.3 | 90.3 | 63.3 | 69.7 | 72.3 | 68.2 | 62.7 | 69.8 |
| | text2vec-base-multilingual | 118M | 71.0 | 66.1 | 33.1 | 78.1 | 43.4 | 59.4 | 81.0 | 62.8 | 63.8 | 67.0 | 66.0 | 55.2 | 62.2 |
| | Angle_BERT | 109M | 77.9 | 76.0 | 37.2 | 75.5 | 45.2 | 68.8 | 85.4 | 64.5 | 66.3 | 70.6 | 67.1 | 57.6 | 66.0 |
| | gte-base | 109M | 74.2 | 91.8 | 46.7 | 85.1 | 48.6 | 86.0 | 93.0 | 72.0 | 71.5 | 76.4 | 71.6 | 57.0 | 73.0 |
| | ALL_862873 | 118M | 50.8 | 52.6 | 22.6 | 36.4 | 22.8 | 50.8 | 61.0 | 29.7 | 34.3 | 44.1 | 54.9 | 40.8 | 41.7 |
| MSE | Student-m | 109M | 76.6 | 89.1 | 44.7 | 87.2 | **60.8** | 88.0 | 95.7 | 81.6 | 77.7 | 82.2 | 67.3 | 60.5 | 76.0 |
| NLL | Student-m | 109M | 79.6 | 89.5 | 45.8 | **88.0** | 59.7 | 88.3 | **96.2** | **83.9** | **78.6** | **82.7** | 67.1 | 61.3 | **76.7** |

### C.2.2 EVALUATION ON SIMILARITY AND CLUSTERING TASKS

**Limited structure of our embedding spaces.** Our method only seeks to pack as much (statistical) information into the embeddings as possible without any constraints on the underlying structure of the embedding space. It is therefore not surprising that methods that relies on metrics on the embedding space such as similarity tasks do not perform as well as the classification tasks. However, our embedder are still competitive on these tasks achieving average performance for their respective size categories.

**Clustering with very small model.** In Tab. 13, we show that our very small model actually outperforms baselines and sits on the pareto frontier for clustering tasks. This is a surprising result as we did not optimize our models for clustering tasks and the embeddings are not designed to have a meaningful structure.

## D VISION

### D.1 TRAINING SET

Tab. 18 presents the statistics, *i.e.* the number of training and testing samples, of the datasets we used for vision. We use the official train sets of the datasets for the knowledge distillation part. We split the official training part to train and validation set with 80 and 20 percents of the data, consequently. The transformation we used on the input image was only a resize transformations to a (225, 225) image. For training the distillation, we extract the embeddings of the train set of each dataset, for each teacher and divide the embeddings to 80 train set and 20 percent validation set.

### D.2 MODEL ARCHITECTURE

The models we used for vision as teachers and student are presented in Tab. 19, including the number of parameters of each of them. For the distillation we use Adam optimizer, with learning rate of 0.001, a batch size of 128, trained for 50 epochs. For fine-tuning for down-stream tasks, we add a two layer fully connected classifier on the frozen embedders, with the first one having the same input dimention as the output dimension, with a leaky ReLU activation function in between.

Table 13: Performance of our distilled models compared of models of similar sizes 16M to 30M parameters from the MTEB Benchmark on clustering tasks.

| | Task Model | Size | Arxiv Clustering P2P | Arxiv Clustering S2S | Reddit Clustering P2P | Reddit Clustering | Stack Exchange Clustering P2P | Stack Exchange Clustering | Twenty Newsgroups Clustering | Avg. |
|---|---|---|---|---|---|---|---|---|---|---|
| MTEB | Bulbasaur | 17M | 40.3 | 31.1 | 51.4 | 45.9 | 30.7 | 52.2 | 39.4 | 41.6 |
| | Ivysaur | 23M | 46.4 | 35.4 | 56.0 | 47.5 | 33.6 | 53.9 | 40.8 | 44.8 |
| | Squirtle | 16M | 33.0 | 24.7 | 43.7 | 31.4 | 29.2 | 39.2 | 28.2 | 32.8 |
| | Venusaur | 16M | 31.8 | 21.1 | 44.1 | 26.7 | 27.5 | 32.8 | 26.1 | 30.0 |
| | Wartortle | 17M | 35.8 | 27.3 | 46.1 | 35.9 | 29.9 | 45.3 | 31.7 | 36.0 |
| | gte-micro | 17M | 35.2 | 31.1 | 47.9 | 45.6 | 30.1 | 52.6 | 40.8 | 40.5 |
| | gte-micro-v4 | 19M | 42.9 | 32.5 | 53.6 | 48.3 | 31.9 | 55.1 | 41.4 | 43.6 |
| | snowflake-arctic-embed-xs | 23M | 43.5 | 32.1 | 57.8 | 48.3 | 34.6 | 57.5 | 36.3 | 44.3 |
| | bge-micro | 17M | 44.6 | 34.5 | 54.5 | 45.3 | 34.7 | 53.1 | 39.4 | 43.7 |
| | bge-micro-v2 | 17M | 44.5 | 33.2 | 55.2 | 45.5 | 34.1 | 54.5 | 40.2 | 43.9 |
| | gte-tiny | 23M | **46.6** | 36.0 | 56.5 | 50.2 | **35.7** | 57.5 | 43.3 | 46.6 |
| | GIST-all-MiniLM-L6-v2 | 23M | 45.3 | 35.5 | 48.7 | 44.1 | 33.9 | 53.1 | 41.1 | 43.1 |
| | slx-v0.1 | 23M | 46.5 | 37.7 | 54.8 | 50.7 | 34.2 | 53.1 | **46.5** | 46.2 |
| | multi-qa-MiniLM-L6-cos-v1 | 23M | 37.8 | 27.7 | 51.0 | 46.3 | 33.4 | 48.1 | 40.8 | 40.7 |
| | all-MiniLM-L6-v2 | 23M | 46.5 | **37.9** | 54.8 | 50.7 | 34.3 | 53.1 | **46.5** | 46.3 |
| | rubert-tiny-turbo | 29M | 24.8 | 16.7 | 40.5 | 26.3 | 28.0 | 33.5 | 19.9 | 27.1 |
| MSE | Student-xs | 23M | 42.4 | 30.9 | 55.2 | 49.2 | 32.7 | 53.5 | 41.9 | 43.7 |
| NLL | Student-xs | 23M | 45.2 | 33.9 | **58.1** | **52.1** | 33.1 | **59.9** | 44.3 | **46.7** |

Table 14: Performance of our distilled models compared of models of similar sizes 30M to 50M parameters from the MTEB Benchmark on clustering tasks.

| | Task Model | Size | Arxiv Clustering P2P | Arxiv Clustering S2S | Reddit Clustering P2P | Reddit Clustering | Stack Exchange Clustering P2P | Stack Exchange Clustering | Twenty Newsgroups Clustering | Avg. |
|---|---|---|---|---|---|---|---|---|---|---|
| MTEB | bge-small-en-v1.5 | 33M | 47.4 | 40.0 | 60.6 | 52.3 | 35.3 | 60.8 | 48.5 | 49.3 |
| | snowflake-arctic-embed-s | 33M | 44.9 | 35.9 | 60.5 | 50.5 | 34.0 | 60.7 | 38.3 | 46.4 |
| | bge-small-4096 | 35M | 43.9 | 29.6 | 54.3 | 43.7 | 33.3 | 51.8 | 36.6 | 41.9 |
| | GIST-small-Embedding-v0 | 33M | 47.6 | 39.9 | 60.6 | 55.5 | 36.2 | 61.9 | **50.0** | 50.2 |
| | NoInstruct-small-Embedding-v0 | 33M | 47.8 | 40.1 | 61.2 | 55.4 | **36.6** | 62.0 | 49.9 | 50.4 |
| | e5-small | 33M | 44.1 | 37.1 | 57.2 | 43.3 | 30.8 | 59.6 | 37.6 | 44.3 |
| | e5-small-v2 | 33M | 42.1 | 34.8 | 59.7 | 45.7 | 32.0 | 58.5 | 41.1 | 44.8 |
| | jina-embedding-s-en-v1 | 35M | 34.2 | 24.0 | 49.9 | 38.0 | 31.5 | 46.4 | 34.4 | 36.9 |
| | jina-embeddings-v2-small-en | 33M | 44.0 | 35.2 | 57.1 | 49.3 | 34.4 | 55.4 | 41.6 | 45.3 |
| | all-MiniLM-L12-v2 | 33M | 46.1 | 37.5 | 54.8 | 51.2 | 33.1 | 53.0 | 47.5 | 46.2 |
| | gte-small | 33M | **47.9** | **40.3** | **61.4** | **55.6** | 36.3 | **62.6** | **50.0** | **50.6** |
| MSE | Student-s | 33M | 43.1 | 33.3 | 57.1 | 50.8 | 32.3 | 55.7 | 42.8 | 45.0 |
| NLL | Student-s | 33M | 45.9 | 35.2 | 60.3 | 51.9 | 32.3 | 61.5 | 45.1 | 47.4 |

Table 15: Performance of our distilled models compared of models of similar sizes 16M to 30M parameters from the MTEB Benchmark on STS tasks.

| Task | Model | Size | BIOSSES | SICK-R | STS12 | STS13 | STS14 | STS15 | STS16 | STS17 | STS22 | STSBenchmark | Avg. |
|---|---|---|---|---|---|---|---|---|---|---|---|---|---|
| MTEB | Bulbasaur | 17M | 85.0 | 76.0 | 69.5 | 81.0 | 77.1 | 85.4 | 82.3 | 88.0 | 64.1 | 83.3 | 79.2 |
| | Ivysaur | 23M | **87.3** | 75.6 | 68.6 | 80.5 | 77.6 | 86.2 | 82.8 | 88.6 | 67.4 | 84.2 | 79.9 |
| | Squirtle | 16M | 71.8 | 77.3 | 70.2 | 78.4 | 74.8 | 82.0 | 78.3 | 85.8 | 61.2 | 79.2 | 75.9 |
| | Venusaur | 16M | 77.6 | 74.7 | 54.4 | 74.2 | 70.0 | 75.7 | 73.7 | 84.8 | 62.6 | 76.7 | 72.4 |
| | Wartortle | 17M | 80.8 | 78.2 | **75.2** | 79.3 | 76.6 | 84.7 | 81.4 | 86.6 | 63.4 | 81.8 | 78.8 |
| | snowflake-arctic-embed-xs | 23M | 84.0 | 69.3 | 65.9 | 77.9 | 72.8 | 83.5 | 80.6 | 84.5 | 66.3 | 79.2 | 76.4 |
| | bge-micro | 17M | 83.4 | 72.4 | 71.9 | 80.9 | 76.6 | 84.9 | 80.7 | 85.6 | 65.9 | 81.3 | 78.4 |
| | bge-micro-v2 | 17M | 82.9 | 73.6 | 71.9 | 79.8 | 76.9 | 84.8 | 81.9 | 86.8 | 65.4 | 82.5 | 78.7 |
| | gte-tiny | 23M | 86.6 | 75.8 | 72.6 | 82.4 | 78.0 | 86.5 | **83.3** | 88.3 | 66.7 | 84.4 | 80.5 |
| | GIST-all-MiniLM-L6-v2 | 23M | 81.3 | 79.1 | 75.0 | **83.3** | **78.6** | **87.0** | 83.0 | 87.4 | **68.1** | 84.4 | **80.7** |
| | multi-qa-MiniLM-L6-cos-v1 | 23M | 79.8 | 70.0 | 64.4 | 76.4 | 69.3 | 80.2 | 79.6 | 81.2 | 65.5 | 76.0 | 74.2 |
| | all-MiniLM-L6-v2 | 23M | 81.6 | 77.6 | 72.4 | 80.6 | 75.6 | 85.4 | 79.0 | 87.6 | 67.2 | 82.0 | 78.9 |
| MSE | Student-xs | 23M | 76.8 | **79.2** | 72.2 | 80.3 | 75.9 | 85.0 | 83.0 | 87.1 | 66.4 | 82.9 | 78.9 |
| NLL | Student-xs | 23M | 78.8 | 77.8 | 71.6 | 80.2 | 77.0 | 85.8 | 82.8 | **89.3** | 65.8 | 83.5 | 79.3 |

Table 16: Performance of our distilled models compared of models of similar sizes 30M to 50M parameters from the MTEB Benchmark on STS tasks.

| Task | Model | Size | BIOSSES | SICK-R | STS12 | STS13 | STS14 | STS15 | STS16 | STS17 | STS22 | STSBenchmark | Avg. |
|---|---|---|---|---|---|---|---|---|---|---|---|---|---|
| MTEB | bge-small-en-v1.5 | 33M | 83.8 | 79.4 | **77.4** | 83.0 | 81.8 | 87.3 | 84.9 | 87.2 | 65.3 | 85.9 | 81.6 |
| | snowflake-arctic-embed-s | 33M | 86.3 | 69.7 | 68.8 | 79.6 | 75.6 | 84.6 | 82.4 | 86.7 | **69.5** | 81.2 | 78.4 |
| | bge-small-4096 | 35M | 81.6 | 74.2 | 72.2 | 80.5 | 76.2 | 85.2 | 81.9 | 86.6 | 65.5 | 81.9 | 78.6 |
| | GIST-small-Embedding-v0 | 33M | 87.0 | **80.5** | 75.6 | **86.3** | **82.3** | **88.7** | **85.3** | **89.0** | 68.5 | **87.1** | **83.0** |
| | NoInstruct-small-Embedding-v0 | 33M | 87.2 | 80.3 | 75.8 | 86.1 | 82.3 | **88.9** | 85.2 | 88.7 | 68.5 | 87.0 | 83.0 |
| | e5-small | 33M | 84.2 | 78.9 | 75.2 | 81.8 | 78.5 | 87.5 | 84.6 | 87.9 | 63.8 | 86.4 | 80.9 |
| | e5-small-v2 | 33M | 79.4 | 78.5 | 76.2 | 82.4 | 79.0 | 87.8 | 83.8 | 87.7 | 63.1 | 86.0 | 80.4 |
| | jina-embedding-s-en-v1 | 35M | 83.0 | 76.3 | 74.3 | 78.5 | 73.8 | 83.7 | 80.0 | 87.5 | 64.2 | 79.2 | 78.1 |
| | jina-embeddings-v2-small-en | 33M | 80.5 | 76.7 | 73.7 | 83.3 | 79.2 | 87.3 | 83.6 | 88.2 | 63.5 | 84.0 | 80.0 |
| | all-MiniLM-L12-v2 | 33M | 83.6 | 79.3 | 73.1 | 82.1 | 76.7 | 85.6 | 80.2 | 88.6 | 65.7 | 83.1 | 79.8 |
| | gte-small | 33M | **88.2** | 77.9 | 75.1 | 85.1 | 81.0 | 88.3 | 83.9 | 87.6 | 68.0 | 85.6 | 82.1 |
| MSE | Student-s | 33M | 78.9 | 79.5 | 70.6 | 79.7 | 75.4 | 84.1 | 81.8 | 86.7 | 66.6 | 83.1 | 78.6 |
| NLL | Student-s | 33M | 81.5 | 79.3 | 73.0 | 81.4 | 78.2 | 86.3 | 84.2 | **90.0** | 66.0 | 84.8 | 80.5 |

Table 17: Performance of our distilled models compared of models of similar sizes 100M to 120M parameters from the MTEB Benchmark on STS tasks.

| | Task / Model | Size | BIOSSES | SICK-R | STS12 | STS13 | STS14 | STS15 | STS16 | STS17 | STS22 | STSBenchmark | Avg. |
|---|---|---|---|---|---|---|---|---|---|---|---|---|---|
| | bge-base-en-v1.5 | 109M | 86.9 | 80.3 | 78.0 | 84.2 | 82.3 | 88.0 | 85.5 | 86.4 | 66.0 | 86.4 | 82.4 |
| | bilingual-embedding-small | 118M | 84.0 | 74.7 | 79.4 | 85.3 | 83.9 | 88.5 | 84.4 | 85.8 | 67.2 | 86.1 | 81.9 |
| | multilingual-e5-small | 118M | 82.3 | 77.5 | 76.6 | 77.0 | 75.5 | 87.1 | 83.6 | 86.4 | 60.9 | 84.0 | 79.1 |
| | snowflake-arctic-embed-m | 109M | 86.6 | 69.1 | 67.0 | 79.1 | 68.5 | 79.9 | 78.7 | 81.5 | 65.8 | 74.1 | 75.0 |
| | snowflake-arctic-embed-m-v1.5 | 109M | 86.4 | 69.9 | 61.8 | 82.7 | 69.0 | 75.5 | 77.3 | 75.0 | 69.1 | 69.7 | 73.6 |
| | GIST-Embedding-v0 | 109M | 88.0 | 81.3 | 76.2 | 87.8 | 83.4 | 89.4 | 85.3 | 88.6 | 67.8 | 87.3 | 83.5 |
| | ml-nlp-elser.html | 110M | 83.8 | 68.8 | 64.8 | 80.1 | 75.0 | 83.7 | 80.5 | 85.7 | 67.5 | 79.5 | 76.9 |
| | e5-base-4k | 112M | 81.4 | 78.3 | 75.8 | 83.6 | 80.0 | 88.8 | 84.5 | 87.6 | 64.1 | 86.5 | 81.0 |
| | instructor-base | 110M | 82.3 | 80.3 | 77.0 | 86.6 | 81.3 | 88.2 | 84.9 | 89.5 | 66.5 | 86.4 | 82.3 |
| | bert-base-uncased | 110M | 54.7 | 58.6 | 30.9 | 59.9 | 47.7 | 60.3 | 63.7 | 64.1 | 56.4 | 47.3 | 54.4 |
| | e5-base | 109M | 85.1 | 79.7 | 74.2 | 83.3 | 78.5 | 88.3 | 84.2 | 87.2 | 62.9 | 86.2 | 81.0 |
| | e5-base-v2 | 110M | 81.4 | 78.3 | 75.8 | 83.6 | 80.0 | 88.8 | 84.5 | 87.6 | 64.1 | 86.5 | 81.0 |
| MTEB | jina-embedding-b-en-v1 | 110M | 83.6 | 79.1 | 75.1 | 80.9 | 76.1 | 85.5 | 81.2 | 89.0 | 66.2 | 82.6 | 79.9 |
| | contriever-base-msmarco | 110M | 83.3 | 70.2 | 64.3 | 80.0 | 74.5 | 83.3 | 79.7 | 86.3 | 64.6 | 78.8 | 76.5 |
| | sup-simcse-bert-base-uncased | 110M | 68.4 | 80.8 | 75.3 | 84.7 | 80.2 | 85.4 | 80.8 | 89.4 | 62.0 | 84.2 | 79.1 |
| | unsup-simcse-bert-base-uncased | 110M | 72.3 | 72.2 | 66.0 | 81.5 | 73.6 | 79.7 | 78.1 | 83.6 | 59.6 | 76.5 | 74.3 |
| | all-mpnet-base-v2 | 110M | 80.4 | 80.6 | 72.6 | 83.5 | 78.0 | 85.7 | 80.0 | 90.6 | 68.0 | 83.4 | 80.3 |
| | allenai-specter | 110M | 65.0 | 56.4 | 62.5 | 58.7 | 54.9 | 62.5 | 64.3 | 69.6 | 55.1 | 61.3 | 61.0 |
| | gtr-t5-base | 110M | 79.0 | 71.5 | 68.6 | 79.1 | 74.6 | 84.8 | 81.6 | 85.8 | 66.2 | 79.6 | 77.1 |
| | msmarco-bert-co-condensor | 110M | 77.3 | 74.0 | 68.2 | 80.4 | 74.0 | 82.6 | 79.8 | 85.9 | 67.5 | 77.0 | 76.5 |
| | paraphrase-multilingual-MiniLM-L12-v2 | 118M | 74.2 | 79.6 | 76.0 | 80.7 | 78.8 | 85.8 | 81.0 | 86.9 | 62.1 | 84.4 | 79.0 |
| | sentence-t5-base | 110M | 75.9 | 80.2 | 78.0 | 85.8 | 82.2 | 87.5 | 84.0 | 89.6 | 62.7 | 85.5 | 81.1 |
| | text2vec-base-multilingual | 118M | 66.2 | 80.0 | 80.9 | 82.9 | 87.4 | 88.3 | 81.6 | 85.8 | 63.0 | 86.5 | 80.2 |
| | gte-base | 109M | 87.6 | 78.9 | 75.7 | 85.7 | 81.5 | 88.8 | 83.8 | 87.9 | 67.3 | 85.7 | 82.3 |
| | ALL-862873 | 118M | 21.3 | 48.5 | 55.6 | 18.4 | 28.8 | 29.2 | 39.0 | 61.2 | 44.5 | 44.4 | 39.1 |
| MSE | Student-m | 109M | 83.4 | 80.9 | 74.5 | 82.8 | 79.0 | 86.6 | 85.2 | 88.4 | 66.4 | 85.2 | 81.2 |
| NLL | Student-m | 109M | 85.2 | 80.2 | 75.2 | 83.4 | 80.4 | 88.3 | 86.0 | 89.9 | 66.2 | 86.4 | 82.1 |

Table 18: Number of training and testing samples in each vision dataset

| Dataset | Number of training samples | Number of test samples |
|---|---|---|
| CIFAR10 Krizhevsky et al. (2009) | 50000 | 10000 |
| FashionMNIST Xiao et al. (2017) | 60000 | 10000 |
| MNIST Deng (2012) | 60000 | 10000 |
| STL10 Coates et al. (2011) | 5000 | 8000 |
| CelebA Liu et al. (2015) | 162770 | 19962 |
| SVHN Netzer et al. (2011) | 73257 | 26032 |
| QMNIST Yadav & Bottou (2019) | 60000 | 60000 |
| KMNIST Clanuwat et al. (2018) | 60000 | 10000 |

Table 19: Number of parameters for each model (in million parameters)

| Model | Number of |
|---|---|
| Swin (Liu et al., 2021b) | 87.77M |
| DINOv2 (Oquab et al., 2023) | 86.58M |
| ViT (Dosovitskiy et al., 2021) | 86.57M |
| BEiT (Bao et al., 2022) | 86.53M |
| PVTv2 (Wang et al., 2022b) | 3.67M |
| WideResNet Zagoruyko & Komodakis (2017) | 68.88M |
| DenseNet Huang et al. (2017) | 28.68M |
| ResNext Xie et al. (2017) | 25.03M |
| ResNet18 He et al. (2016) | 11.69M |
| GoogLeNet Szegedy et al. (2015) | 6.62M |
| MNASNet Tan et al. (2019) | 4.38M |
| MobileNet Sandler et al. (2018) | 3.50M |
| ShuffleNet Ma et al. (2018) | 2.28M |
| SqueezeNet Iandola et al. (2016) | 1.25M |

We use SGD optimizer, with a learning rate of $0.001$, L2 penalty of $0.0001$, a momentum of $0.9$, with Nesterov momentum enabled, and a batch size of 64.

### D.3    COMPLEMENTARY RESULTS

Considering the limited space, we gather all the experiment for all possible student architectures in Tab. 20. As shown in the table, for all the possible student architecture, our method outperforms the other multi-teacher feature distillation methods, and all the teachers, in classification of all datasets, except for STL10 dataset. For STL10, we can see that it outperforms other multi-teacher feature-distillation methods in general. Also, Figure 14 illustrates how our method outperforms other distillation methods as well as the non-distilled teachers, for all but one architecture (squeezenet), demonstrating the significant improvement achieved compared to other distillation baselines.

Furthermore, you can see the detailed comparison of our multi-teacher feature distillation, with its single-teacher version in Tab. 21 for all possible teachers, with resnet18 as the student. Again except for STL10, our method outperforms the single-teacher case, with being the second best for STL10. Tab. 22 also shows the detailed results of the second setting of vision modality, i.e. the Vision Transformer teachers and students.

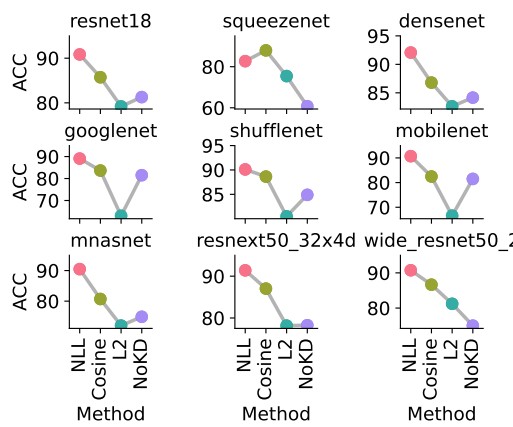

Figure 14: Comparison of accuracy of our method (NLL), no distillation teachers, and other distillation methods (L2 and Cosine), across different student architectures and tasks.

Table 20: Distillation results for each nine model as the student, distilled with NLL, Cosine, and MSE, compared with their fine-tuning performance without distillation, for seven tasks.

| Method | Model | CIFAR10 | FMNIST | MNIST | STL10 | SVHN | QMNIST | KMNIST |
|--------|-------|---------|--------|-------|-------|------|--------|--------|
| NoKD | resnet18 | 78.01 | 87.02 | 96.71 | 92.26 | 38.43 | 96.60 | 79.97 |
| | squeezenet | 52.96 | 66.47 | 69.37 | 51.06 | 34.95 | 78.29 | 72.28 |
| | densenet | 85.45 | 87.99 | 95.55 | **97.06** | 48.44 | 95.44 | 79.17 |
| | googlenet | 80.20 | 85.40 | 95.50 | 93.52 | 44.03 | 95.50 | 76.50 |
| | shufflenet | 82.63 | 88.35 | 97.22 | 91.31 | 52.28 | 97.12 | 85.37 |
| | mobilenet | 77.23 | 85.90 | 96.06 | 92.47 | 43.44 | 96.08 | 79.23 |
| | mnasnet | 76.97 | 80.27 | 89.89 | 70.56 | 39.64 | 88.63 | 78.25 |
| | resnext50-32x4d | 79.78 | 84.31 | 92.54 | _94.94_ | 38.11 | 92.34 | 65.86 |
| | wide-resnet50-2 | 77.56 | 81.78 | 87.74 | 94.44 | 34.83 | 86.83 | 61.89 |
| Cosine | resnet18 | 84.82 | 90.01 | 98.86 | 86.50 | 53.03 | 98.75 | 88.12 |
| | squeezenet | 84.16 | 90.83 | 98.98 | 86.98 | 63.74 | 98.93 | 92.00 |
| | densenet | 86.60 | 90.53 | 98.82 | 88.54 | 54.99 | 98.80 | 89.28 |
| | googlenet | 83.42 | 88.79 | 98.16 | 77.88 | 56.36 | 98.28 | 82.57 |
| | shufflenet | 86.17 | 91.06 | 98.41 | 85.41 | 69.74 | 98.55 | 91.14 |
| | mobilenet | 83.74 | 83.79 | 98.38 | 81.55 | 50.90 | 98.54 | 80.32 |
| | mnasnet | 84.06 | 89.06 | 98.06 | 59.22 | 50.00 | 98.18 | 86.35 |
| | resnext50-32x4d | 86.04 | 84.82 | 98.65 | 86.94 | 66.98 | 98.54 | 87.22 |
| | wide-resnet50-2 | 86.00 | 89.79 | 98.46 | 86.26 | 61.89 | 98.49 | 85.63 |
| L2 | resnet18 | 81.70 | 85.10 | 96.88 | 74.31 | 40.06 | 96.60 | 79.75 |
| | squeezenet | 76.78 | 83.26 | 89.42 | 64.47 | 43.24 | 97.60 | 73.28 |
| | densenet | 84.21 | 84.31 | 97.81 | 84.47 | 47.87 | 97.59 | 82.15 |
| | googlenet | 65.60 | 81.70 | 88.28 | 13.75 | 38.06 | 94.52 | 59.22 |
| | shufflenet | 81.87 | 87.76 | 97.55 | 58.74 | 58.74 | 97.53 | 81.20 |
| | mobilenet | 74.68 | 82.97 | 79.28 | 34.24 | 39.66 | 88.73 | 66.39 |
| | mnasnet | 71.57 | 82.99 | 89.03 | 47.64 | 39.36 | 96.93 | 76.61 |
| | resnext50-32x4d | 82.49 | 84.23 | 97.09 | 60.31 | 42.81 | 97.88 | 82.36 |
| | wide-resnet50-2 | 82.87 | 84.40 | 98.29 | 76.42 | 45.08 | 98.07 | 83.37 |
| NLL | resnet18 | 86.09 | 91.38 | 99.15 | 86.05 | 83.33 | **99.15** | 90.75 |
| | squeezenet | 70.74 | 83.59 | **99.21** | 70.46 | 62.06 | 98.93 | **93.86** |
| | densenet | **88.07** | _91.75_ | _99.17_ | 88.60 | **85.15** | 99.13 | 92.65 |
| | googlenet | 85.95 | 90.50 | 98.97 | 85.94 | 73.03 | 99.04 | 90.16 |
| | shufflenet | _87.66_ | **91.95** | 98.85 | 87.02 | 73.48 | 98.93 | _92.85_ |
| | mobilenet | 86.85 | 91.64 | 99.01 | 86.49 | 79.42 | 99.01 | 92.48 |
| | mnasnet | 87.55 | 91.39 | 98.88 | 87.60 | 78.16 | 98.85 | 90.88 |
| | resnext50-32x4d | 87.20 | 91.70 | 99.10 | 87.35 | _84.32_ | 99.03 | 91.06 |
| | wide-resnet50-2 | 86.71 | 91.01 | 98.87 | 85.99 | 82.89 | 98.95 | 90.76 |

Table 21: Comparison of single-teacher scenario with multi-teacher one for resnet18 as the student.

| Method | Model | CIFAR10 | FMNIST | MNIST | STL10 | SVHN | QMNIST | KMNIST |
|--------|-------|---------|--------|-------|-------|------|--------|--------|
| | Multi-teacher | **86.09** | **91.38** | **99.15** | 86.05 | **83.33** | **99.15** | **90.75** |
| NLL | squeezenet | 77.59 | 88.78 | 97.98 | 77.88 | 56.64 | 97.87 | 85.95 |
| | densenet | **86.09** | **90.80** | 98.46 | **86.66** | _73.37_ | 98.50 | 89.45 |
| | googlenet | 82.92 | 89.74 | 98.54 | 85.30 | 68.23 | 98.41 | 88.31 |
| | shufflenet | 79.54 | 90.38 | _98.68_ | 79.50 | 67.56 | 98.61 | 90.09 |
| | mobilenet | 78.88 | 89.95 | _98.68_ | 80.49 | 66.43 | _98.57_ | 88.90 |
| | resnext50-32x4d | 81.67 | 90.39 | 98.47 | 82.40 | 68.57 | 98.30 | 87.66 |
| | wide-resnet50-2 | 81.50 | 90.19 | 98.59 | 82.73 | 67.63 | 98.40 | 87.40 |
| | mnasnet | 81.20 | 90.35 | 98.52 | 81.94 | 65.93 | 98.52 | _90.29_ |

Table 22: Comparison of Vision Transformer teachers and students for second setting of vision.

| Method | Model | Parameters | CIFAR10 | DTD | STL10 | SVHN | FGVCAircraft | CUB |
|--------|-------|------------|---------|-----|-------|------|--------------|-----|
| NoKD | Swin | 87.77M | 97.67 | 75.80 | **99.60** | 56.70 | 38.58 | 78.01 |
| | ViT | 86.57M | 96.90 | 70.59 | 99.40 | 50.14 | 33.60 | 65.65 |
| | DINOv2 | 86.58M | **98.57** | **80.64** | 99.45 | 57.30 | 30.60 | **81.88** |
| | BEiT | 86.53M | 97.89 | 75.27 | **99.60** | 62.00 | **49.59** | 23.21 |
| | PVTv2 | 3.67M | 88.70 | 63.67 | 95.72 | 62.20 | 25.68 | 39.96 |
| | resnet18 | 11.69M | 76.76 | 47.18 | 87.19 | 54.66 | 26.25 | 33.19 |
| NLL | PVTv2 | 3.67M | 94.76 | 61.33 | 96.51 | **77.87** | 40.35 | 56.99 |
| | resnet18 | 11.69M | 95.21 | 46.38 | 94.86 | 77.58 | 32.34 | 23.39 |

# E   COMPUTATIONAL COST AND COMPLEXITIY

**Teachers' embeddings.**   To reduce the computational cost we first embedded the entirety of the training set using the teachers and store them. We can then build training batches by sampling from the pre-computed embeddings. In NLP this amounts to around to a total of 91GB of embeddings for our 4 teachers.

**Hardware.**   We trained our models on NVIDIA A100 GPUs with 80GB of memory. All our models were trained on a single GPU using pytorch and pytorch lightning.

**Time complexity.**   For our molecular experiments, training on the largest dataset took two days, 5 hours on the ZINC-250k datasets, one day for computer vision and 8 days in NLP. We display in Figure 15 the evolution of the runtime of one step with a batch size of 256 with our molecular embedders (computed over 10 runs). The complexity of our algorithm is linear with the number of teachers, and an additional teacher increases the runtime of one training step by 1.57 ms, representing less than 1% of the total runtime.

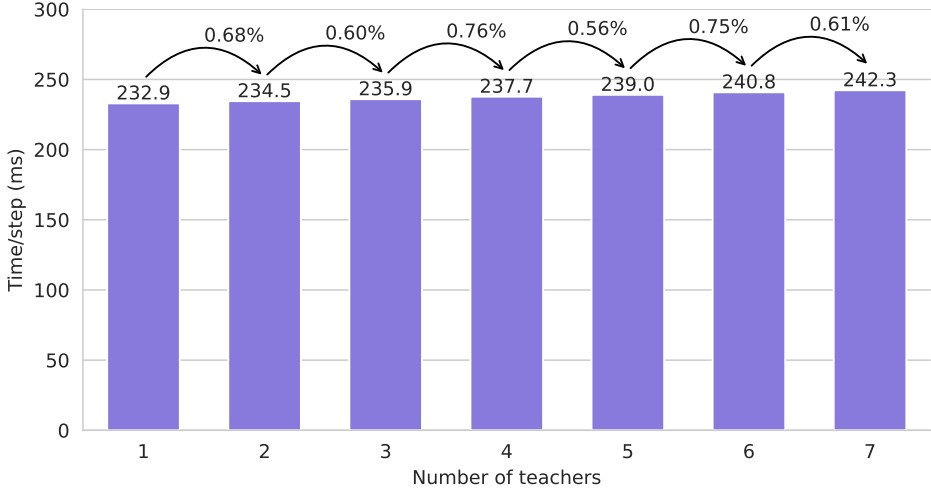

Figure 15: Evolution of the time to perform one training step with a batch size of 256 in molecular modeling. The computational overhead induced by an additional teacher represents less than 1% of the total runtime on a batch.

# F    DETAILED METHOD

---

**Algorithm 1** Distillation through Gaussian Kernels

---

**Input:** Dataset $D = \{\mathbf{x}_i\}$, Embedders $(\mathsf{T}_k)_{1 \leqslant k \leqslant K}$, Student embedder $\mathsf{S}$, Number of iterations $T$, Learning rate $\eta$

Initialize the parameters $\theta_s$ of the student embedder $E_s$ and the parameters $\theta_k$ of the parametric Gaussian kernels

**for** $t = 1$ to $T$ **do**

    Sample a batch of inputs $\{\mathbf{x}_i\}$

    Compute the embeddings $\left\{ \mathbf{t}_i^k = \mathsf{T}_k(\mathbf{x}_i) \right\}_{1 \leqslant k \leqslant K}$

    Compute the student embeddings $\{\mathbf{s}_i = \mathsf{S}(\mathbf{x}_i)\}$

    Compute the loss $\mathcal{L}_{NLL} = -\sum_{k=1}^{K} \sum_{i=1}^{N} \log \mathcal{N}(\mathbf{t}_i^k | \mu_k(\mathbf{s}_i), \Sigma_k(\mathbf{s}_i))$

    Update the parameters $\theta_s$ and $\theta_k$ using the Adam optimizer.

**end for**

---

# G    BASELINES

For the MSE, we will optimize the following loss function:

$$\mathcal{L}_{MSE} = -\sum_{k=1}^{K} \sum_{i=1}^{N} ||\mathsf{S}(\mathbf{x}_i) - \mathsf{T}_k(\mathbf{x}_i)||^2 , \tag{8}$$

where it calculates the summation of L2 distances between the representation produced by each teacher and the student, for each instance of the batch.

For Cosine multi-teacher feature distillation, we optimize the summation of cosine of teachers and the students representations of each instance of the batch, *i.e.*:

$$\mathcal{L}_{Cosine} = -\sum_{k=1}^{K} \sum_{i=1}^{N} \frac{\mathsf{S}(\mathbf{x}_i).\mathsf{T}_k(\mathbf{x}_i)}{\max(||\mathsf{S}(\mathbf{x}_i)||_2 . ||\mathsf{T}_k(\mathbf{x}_i)||_2 , \epsilon)}. \tag{9}$$

