# OpenReview forum: "How to distill task-agnostic representations from many teachers?"
_ICLR.cc/2025/Conference — Submitted to ICLR 2025_

### Official Review · Reviewer_yAYt · 2024-10-18

**Soundness:** 3
**Presentation:** 3
**Contribution:** 3
**Rating:** 6
**Confidence:** 4

**Summary:**

This paper proposes a simple and effective method for distilling task-agnostic representations from an ensemble of teacher models. The authors frame the multi-teacher distillation problem as a task-enabling problem and introduce a novel approach using Gaussian kernels to estimate the conditional distribution of teacher embeddings given the student embedding.

**Strengths:**

1) Simplicity and Effectiveness: The proposed method is remarkably simple and yet achieves surprisingly strong results on various tasks. This is a significant strength of the paper.

2) Strong Empirical Gains: The experimental results demonstrate clear empirical gains over existing methods, showcasing the effectiveness of the proposed approach.

3) Comprehensive Evaluation: The authors evaluate their method on a diverse set of tasks, including image classification, text classification, demonstrating the versatility of the distillation technique.

**Weaknesses:**

1) Metric Space Preservation: As acknowledged by the authors, the proposed method might not always preserve the metric space of the teacher embeddings, depending on the task at hand. This could be a potential limitation but not something I would hold against the paper -- sometimes things might work better for a specific aspect, it would be nice to investigate why or how to fix things given normal embedding models are generalists for this one reason.

2) Limited Model and Data Scale: The experiments are primarily conducted with relatively small models and datasets. It would be more compelling to see results with larger models for both vision and language and larger datasets to assess the scalability of the method.

**Questions:**

This paper presents an interesting and effective method for multi-teacher distillation with strong empirical results. However, the potential issue with metric space preservation and the limited exploration of larger models and datasets slightly weaken the current submission.

Recommendation:

I am leaning towards acceptance, but I believe the paper could be strengthened by addressing the following points:

1) Provide further analysis and discussion on the potential impact of metric space distortion on different tasks.
2) Consider experimenting with larger models and datasets to demonstrate the scalability of the method.

I am interested in seeing other reviewers' thoughts on these points during the discussion phase.

---

> ### Author Response · Authors · 2024-11-19
>
> We thank Reviewer yAYt for their detailed review and the effort they put into evaluating our work.
>
> ## Questions
>
> 1) **We extended the discussion on the metric space distortion (see Sec 6.2: Embedding space structure), and moved results located in the appendix to the main body of the paper (Fig.5).**
> The metric space distortion stems from the intrinsic properties of the mutual information. The mutual information is a metric that is invariant through composition of invertible mappings (for two invertible maps $f_1$ et $f_2$, $I(X;Y) = I(f_1(X); f_2(Y))$). As a result, this objective cannot preserve structural properties of the teachers (such as the pairwise cosine similarity).
> This invariance is thus a limitation of our method: it only enforces the preservation of information (retrievable by arbitrarily complex schemes or extraction mechanisms). This is because of this “extraction mechanism” requirement that our results for classification tasks hold (we train a classifier to extract the relevant information on top of the embedding) but not for tasks such as clustering or sentence similarity that do not rely on an additional complex extraction mechanism. Regardless, we evaluated our models in both these settings in text. In section C.2.2, we provided the results and analysis of clustering and STS.
>
> 2) **Our experiments were conducted on larger models and datasets in NLP.** We evaluate models from 22 to 109M parameters (we updated the paper to present the results for all three sizes in the body of the paper). We utilized standard architectures commonly employed in practical scenarios, specifically the Snowflake family, we trained on a common large-scale embedding-oriented datasets from hugging faces comprising around 1B tokens. We evaluated the resulting embedders on the standardized MTEB benchmark: comprising 12 classifications datasets, 7 clustering datasets, and 10 STS datasets, all of which are widely accepted and commonly used for embedding evaluation.
>
> We hope these clarifications alleviate your concerns, please let us know if you have any additional questions.

---

> > ### Comment · Reviewer_yAYt · 2024-11-20
> >
> > Thank you. This is helpful. I am positive on the paper but not confident enough to raise the score to 8.

---

### Official Review · Reviewer_qU1H · 2024-10-30

**Soundness:** 3
**Presentation:** 3
**Contribution:** 3
**Rating:** 6
**Confidence:** 4

**Summary:**

The paper presents a multi-teacher knowledge distillation approach for creating task-agnostic representations, leveraging conditional entropy and Gaussian kernel-based learning. The authors aim to transfer diverse information from multiple pretrained teacher models to a student model, with applications in NLP, vision, and molecular modeling. The method introduces a task-agnostic loss function optimized by minimizing negative log-likelihood, showing improvements over baseline distillation methods (e.g., MSE and cosine similarity) across tasks.

**Strengths:**

1. The authors introduce a task-agnostic knowledge distillation framework, moving beyond traditional task-specific approaches. This is particularly valuable for generalizing representations across diverse tasks.

2. The paper offers a strong theoretical grounding for the proposed task-agnostic distillation loss, including insights into its formulation using conditional entropy and mutual information maximization.

3. The method is evaluated across three domains: vision, molecular modeling, and NLP, demonstrating consistent improvements in each, suggesting general applicability.

4. Results consistently show that the student models trained with this method outperform those trained with traditional distillation methods in accuracy, robustness, and generalizability.

**Weaknesses:**

1. The paper lacks a detailed discussion of potential challenges, such as computational overhead in training with multiple teacher models or the scalability of the method to even larger models and datasets.

2. While the approach is tested against traditional distillation baselines, comparing with recent task-agnostic distillation methods, if any, would strengthen the empirical claims.

3. Although theoretically task-agnostic, the method’s robustness across a more extensive range of tasks within each domain is less explored. Additional benchmarks, particularly in NLP, could solidify the approach’s adaptability.

**Questions:**

1. It is recommended to clarify how the task-agnostic nature of the proposed approach is ensured. For example, does this approach impose any constraints to PREVENT the method from IMPLICITLY learning task-specific nuances from the data?

2. How does the task-agnostic method generalize across tasks of a highly varying nature (e.g., between NLP, vision, and molecular modeling)? Are there tasks where performance might degrade without task-specific fine-tuning? Discussing them more will enhance the expressiveness of the paper.

3. The paper discusses the computational efficiency of the approach. It is suggested to provide additional details or empirical results on the computational cost, particularly when applying this method to large datasets or models.

4. How does the computational cost of this method compare to other baseline or existing SOTA task-agnostic and task-specific distillation techniques?

5. Is there a specific reason for choosing Gaussian kernel-based learning over alternative distance metrics for teacher-student alignment?

6. Although the method is compared to standard baselines, it is highly recommended to provide a comparison against recent task-agnostic distillation methods, if any exist. A comparative analysis would provide a clearer view of the approach’s relative advantages.

6. Please clarify if this approach uses different hyperparameters for the three domains (NLP, vision, molecular modeling) or maintains a consistent configuration across tasks.

7. The empirical evaluation seems broad, but how does the model handle fine-grained tasks or those with limited training data? For instance, is it still robust in few-shot learning scenarios?

8. Does the proposed method require specific architectures for the teacher and student models? Would the method be effective if applied to architectures like transformers versus CNNs?

---

> ### Author Response · Authors · 2024-11-19
>
> We would like to thank Reviewer qU1H for their detailed account of our work and the efforts they put into their review.
>
> ## Weaknesses
> 1) **We added details to the Section E (“Computational cost and Complexity”) to show the effect of the number of teachers on the runtime of the training**, showing that an additional teacher increases the runtime by less than 1%.
> **Our method is scalable**, in text, we specifically trained models up to 100m parameters with limited resources and got good performance. (See Table 3, Fig 6). We trained on 5M text samples in text (on a SOTA/modern dataset: GIST) and on 2M samples for molecular embedding. Overall, we trained on SOTA realistic settings both in terms of model sizes and dataset sizes in text and molecular data releasing SOTA models. We edited the main table of the NLP section to reflect the results on the three model sizes that were previously relegated to the appendices.
>
> 2) **In the multi-teacher setting, to the best of our knowledge, no method covers the task agnostic distillation framework.** Most of the single teacher approaches cannot be directly adapted to this setting, we therefore compared our method to two baselines that could easily be adapted to the multi-teacher setting: distillations based on the cosine similarity and the L2 distance.
>
> 3) **We evaluated a total of 33 datasets in text.** Encompassing classification tasks, clustering tasks and STS tasks. Our theoretical results hold only on classification tasks thus these are the main results reported in the body of the paper but we provide results in clustering and STS in appendices (See Fig 6).
> Furthermore, we also evaluated our method on 32 datasets in molecular modeling, covering both regressions and classification tasks.
>
>
> ## Questions
>
> 1) Our method is task-agnostic in the sense that it does not require models to be trained on specific tasks nor labels.
> While training the student, only views of the same datapoint through different teachers is required, building  a single representation that encompasses as much information about all other representations.
>
> 2) We added a discussion and a visualization to discuss the robustness of our method on tasks containing varying amounts of samples, and for imbalanced classification datasets (Sec.B.5).
> We observed that while remaining competitive, as the datasets become more imbalanced, or as the amount of available sample decreases, the performance of our students and the top performing teachers become more similar.
> Finally, there are tasks where our student’s performance are not as competitive, for instance in molecular modeling, on ClinTox, our model achieves an accuracy of 66% in our experiments while in this same experimental setup, ThreeDInfomax achieves an accuracy of 84%. Here, it can be explained by the fact that ClinTox contains drugs that have gone through clinical tests, while our training set focuses on lead molecules, most of which have not passed the pre-clinical phase.
>
> 3) **We extended the discussion on computational cost and included additional empirical results in Sec.E.** Notably we provide a visualization of the computational overhead induced by the addition of multiple teachers to our distillation process.
>
> 4) Compared to other distillation techniques, our approach performs similarly in terms of time complexity. Indeed, the estimation of Gaussian Mixtures does not require significantly more parameters than a simple dense cross encoder used in previous SOTA baselines using an L2 loss or a Cosine Similarity, and these cross-encoders are significantly smaller than the backbone of the student. As a result for all of these methods, the principal bottleneck in terms of runtime is the backbone of the student (see Fig.14 in Sec.E).
>
> 5) We rely on the theoretical results on mutual information and conditional entropy estimation methods proposed in KNIFE [1]. **Those results would not hold for an arbitrary class of function [2]**. Mixtures of Gaussian distributions are dense in the set of probability distributions, with respect to the weak topology. By "weak topology" we mean the probabilists' weak topology, also called the topology of convergence in distribution, the vague topology, and the weak-* topology. In general, under smoothness conditions, people perform "density estimation" with Gaussian "kernel functions". I.e. convolve the empirical distribution (the finite discrete distribution of the points in your sample) with a Gaussian density. We try to approximate the unknown population density with a weighted average of Gaussians.
>
>
>
> [1] Georg Pichler, Pierre Colombo, Malik Boudiaf, G¨unther Koliander, and Pablo Piantanida. A differential entropy estimator for training neural networks, 2022
>
> [2] Donoho, David L. “One-Sided Inference about Functionals of a Density.” The Annals of Statistics 16, no. 4 (1988): 1390–1420.

---

> > ### Author Response · Authors · 2024-11-19
> >
> > ## Questions
> >
> > 6) We compare our methods to different pretrained embedders, but we identified no other method covering both the multi-teacher and the task-agnostic setting. We therefore adapted two common task-specific baselines that could easily be adapted to our setting.
> > Furthermore, in molecular modeling, we compare our students to two baselines using a contrastive approach in its pre-training phase: ThreeDInfomax and GraphMVP, both aiming at incorporating 3D information into 2D models.
> >
> > 7) The only hyperparameter that our method introduces is the architecture/dimension of the encoders used to learn the conditional Gaussian distributions. We used the same 2 layers feedforward network of hidden dimensions of the student's output and predicted a mean vector and a diagonal covariance matrix. For the training we used the same standard parameters (Adam lr=10^-5 ).
> >
> > 8) In the revised version of the paper, Fig.13 shows in molecular modeling the evolution of our model’s ranking with the amount of samples in a dataset. **We extended our result section in vision with the datasets FGVC-Aircraft and DTD for texture classification to cover fine-grained classification tasks**.
> >
> > 9) **The method is architecture agnostic.** Our method can handle students and teachers of any architecture and embedding dimensions since it is only a specific loss. In the molecular embedding experiments, the students are trained with teachers with a wide range of architectures, from 2D and 3D graph neural networks to transformers, with varying embedding dimensions.
> >
> >
> > We renew our thanks to Reviewer qU1H for their thorough review of our work, and we hope our answers addressed all their concerns so they can re-assess our work’s quality.

---

> > > ### Author Response · Authors · 2024-11-29
> > >
> > > Dear Reviewer qU1H,
> > >
> > > We hope this message finds you well. While the revision period has concluded, we remain available to address any questions or provide clarification regarding our rebuttal. Your feedback is invaluable to us, and we would greatly appreciate any additional comments you may have.
> > >
> > > Thank you for your time and consideration.

---

> > > > ### Comment · Reviewer_qU1H · 2024-12-02
> > > > **Final Rating**
> > > >
> > > > Thanks to the authors for the reply. I appreciate the responses. I plan to keep my score.

---

### Official Review · Reviewer_CDoW · 2024-11-02

**Soundness:** 2
**Presentation:** 2
**Contribution:** 3
**Rating:** 5
**Confidence:** 4

**Summary:**

The paper proposes a method for task-agnostic distillation from multiple pretrained teachers.
The problem is formulated as minimizing the disagreement between the student's predictions and the teacher's for any given task, which can be bounded by the conditional entropy of the teacher's embedding given the student's. This upper-bound does not depend on the specific task, and can therefore be chosen as a loss for task-agnostic distillation. The conditional distribution of teacher's embedding given the student's is parametrized by a Gaussian model whose mean and covariance are learned.

The distillation method is evaluated on three different applications:
- **Image classification** on CIFAR10, STL10, SVHN and variants of MNIST, with a ResNet18 as student and 9 teachers ranging in size from SqueezeNet to WideResNet50-2. Task-agnostic distillation is performed on the combination of datasets used for downstream evaluation.
- **Molecular property prediction (regression and classification)** on the ADMET and HTS tasks, with 9 teachers trained on different modalities (textual representations of molecular graphs, 2D-GNN and 3D-GNN) and a 10-layer GINE as the student. Task-agnostic distillation is performed on two datasets (ZINC-250k and ZINC-2M).
- **Natural language processing** on the MTEB benchmark, with four large teachers and the snowflake Merrick models as students. Task-agnostic distillation is performed on 6 million entries from a collection of datasets from the Huggingface Hub.

In these different settings, the teachers are chosen to be diverse in size, and with a performance varying across tasks. The trained student is shown to have consistently good results across all tasks, and to outperform all teachers on average.

**Strengths:**

- The paper tackles the important question of task-agnostic knowledge distillation from multiple teachers.  While task-agnostic knowledge distillation and task-specific multi-teacher distillation have been broadly studied, task-agnostic multi-teacher distillation is still understudied.
- The proposed distillation loss is original and well-motivated.
- The proposed approach is evaluated on diverse applications: classification in 2D vision, molecular property prediction with GNNs, and natural language processing.

**Weaknesses:**

**Related work.**
The related work overlooks a large portion of the literature on task-agnostic distillation. The question has been extensively studied, see e.g, Abbasi Koohpayegani
et al., 2020; Fang et al., 2021; Xu et al., 2022; Gao et al., 2022; Navaneet et al., 2021; Wu et al., 2022; Duval et al., 2023. None of these works require, *finetuning the teacher*, or *jointly training the teacher and the student* as claimed in the related work, and most of them do not require *the teacher and the student to have the same architecture*.

In light of the abundance of works on task-agnostic knowledge distillation, the following claim of the study could be revised:
*To the best of our knowledge, there are few methods that address task-agnostic representation distillation, especially in the context of multi-teacher approaches.*


**Experimental validation.**

Being unfamiliar with the literature on molecular property prediction and natural language processing, my remarks are focusing on the experiments on 2D vision.

1. The evaluation protocol for vision tasks is based on very small datasets (MNIST, STL10, SVHN, CIFAR10) with very small image sizes (MNIST is 28 $\times$ 28, CIFAR10 is 32 $\times$ 32). Using larger, more varied datasets would better showcase the method’s applicability across real-world scenarios. Even if computational resources were a constraint, there are numerous small-scale datasets more relevant than CIFAR or MNIST, such as those from the fine-grained classification community (e.g., CUB bird classification, FGVC-Aircraft, DTD for texture classification).

2. The teachers are small CNN models, which makes it challenging to fully appreciate the contributions of the authors. This study would benefit greatly from evaluations using larger ViT teachers such as DINOv2.

3. Lastly, as there is no data augmentation applied (Appendix D.1 only mentions a resizing to $(225, 225)$), the experimental results are very low. The achieved results are far lower than what could be obtained from training from scratch with proper data augmentation. It is not clear whether in a more challenging scenario with properly trained teachers, the conclusion would still hold. I suggest using standard augmentation strategies for training the teacher and student models, such as color jittering, cutout and random cropping (for CIFAR or MNIST), random resizing and cropping (for larger RGB images).

**Minor remarks.**

- *To do so, we first measure the agreement between the student’s Bayes classifier and the teachers’ for any given task. We show that it can be bounded by the conditional entropy of the teacher’s embedding given the student’s,*
$\rightarrow$ the conditional entropy bounds the **disagreement**, not the **agreement**.

**Questions:**

**Summary of suggestions** - please refer to Weaknesses section
- In the related work, mention the abundant literature on task-agnostic knowledge distillation
- For vision tasks, using **larger pretrained teachers** such as DINOv2, **evaluating on more impactful downstream tasks**, and **training models with appropriate data augmentation**. These experiments would demonstrate the applicability of the proposed approach to real-world scenarios.

---

> ### Author Response · Authors · 2024-11-19
>
> We would like to thank Reviewer CDoW for their detailed account of our work and the efforts they put into their review.
>
>
> ## Weakness: Related work
>
> Thank you for pointing out these additional references. In this related work section, we refer to task-agnostic distillation, that is to say the distillation process was performed without labels, or task-specific loss. We revised the formulation of this section, to include the proposed references. Please note that in the various references proposed, 2 are task specific (the student is trained using a task specific loss). While the other citations do address task-agnostic knowledge distillation, none of them treats the multi-teacher framework. We included those references to our related work section.
>
> While we believe they do not invalidate our claim, as they all rely on a single teacher, we still revised our claim to focus it on the multi-teacher setting: To the best of our knowledge, there are few methods that address task-agnostic representation distillation in the context of multi-teacher approaches.
>
>
> *For a more complete discussion on the proposed references please refer to the following general comment.*
>
> ## Weakness: Experimental Validation
>
> 1,2 ) We have included in the revised version of the article larger models such as **DINOv2**, **Swin**, **BEiT**, **ViT**, and **PVTv2** as teachers, evaluated on **CUB**, **FGVC-Aircraft**, and **DTD**, in addition to CIFAR10, SVHN and STL10.
>
> To present the expanded evaluations, we have added a visualization of the **Pareto frontier between model size and performance** to the main paper (Figure 2). This highlights the trade-offs achieved by our method. Additionally, the full set of results, including the new datasets and teacher models, is provided in Table 22 of the appendix for detailed comparison.
>
> 3 ) To perform downstream tasks, we chose to freeze the model’s backbone and train classifiers using its embeddings. This experimental setup (Sec 3.3) was designed specifically to assess the quality and informativeness of the embeddings we produce. This explains the lower values of all our metrics, compared to experimental setups where the full backbone is fine-tuned.
> We added a remark in our analysis of the results in vision to make sure this information appears clearly in the article.
>
> Finally, we evaluated resnet18 with the suggested augmentation. We observed improvements over certain benchmarks (notably CIFAR10), even though none of them outperform the student without any augmentation. To see the results, please refer to the following table (some experiments are still running, notably the student with data augmentation, we will update the revised article in the following days).
>
> *Embedder accuracy on various benchmarks using data-augmentations:*
>
> |Embedder|CIFAR10|FMNIST|MNIST|STL10|SVHN |QMNIST|KMNIST|
> |-----------|-------|------|-----|-----|-----|------|------|
> |resnet18   |81.89  |86.94 |96.6 |92.98|51.01|96.89 |80.43 |
> |squeezenet |79.23  |86.65 |97.51|85.82|47.77|97.59 |84.05 |
> |googlenet  |81.94  |86.38 |96.71|93.95|55.9 |97.2  |79.27 |
> |shufflenet |81.61  |87.57 |95.77|71.51|49.08|95.96 |76.97 |
> |mobilenet  |81.67  |88.07 |96.05|92.26|48.57|97.5  |  -   |
>
> ## Minor Remark.
> Thank you for the remark, we modified this sentence in the revised document.
>
>
> We extend our thanks to Reviewer CDoW for their thorough review of our work, and we hope our answers and these additional discussions addressed all their concerns so they can reassess our work’s quality.

---

> > ### Author Response · Authors · 2024-11-19
> > **Details on the mentionned references**
> >
> > We thank reviewer CDoW for the extensive references they pointed out. In this comment we will cover how they relate to our work. As explained in the main answer, we found no reference addressing the training of students with task-agnostic multi-teacher knowledge distillation.
> > First we found 2 references who proposed distillation processes that are not task-agnostic:
> > - **Navaneet et al. 2022:** a multi-teacher approach to distill knowledge **on one specific task** (cf eq.3). Nevertheless, we acknowledge this work is related to our L2 distillation baseline which we now reference in our paper.
> > - **Wu et al., 2022:** This work focuses on the distillation of a single teacher and is **task-dependent** (cf line 18 of Algorithm.1).
> >
> > All of the remaining references focus on the distillation of a single teacher, and often make assumptions on the student’s architectures that cannot be enforced in our application. Since these works are related to our method, we include them to our Related work section.
> > - **Koohpayegani et al., 2020:** the proposed method focuses on a single teacher, and yields few requirements on the student architecture ("2-q" model).
> > - **Fang et al. 2021:** a single teacher  approach requiring both the student and teacher to have the same dimension.
> > - **Xu et al., 2022:** This method requires the student and the teacher to have the same architecture, and is applied to a single teacher setting. We found another reference from ICLR’22 sharing similarities with contrastive approaches.
> > - **Gao et al. 2022:** The method is applied to a single teacher setting and uses vision specific data-augmentation.
> > - **Duval et al. 2023:** This method specifically focuses on the distillation of joint embedding approaches, in the single teacher setting.

---

> > > ### Comment · Reviewer_CDoW · 2024-11-23
> > >
> > > I thank the authors for taking the time to integrate the references on task-agnostic knowledge distillation.
> > >
> > > The additional experiments with larger ViT models on tasks beyond CIFAR and MNIST are much welcomed; however, I am still confused by the low reported results. The authors did not specify which DINOv2 model they are using (ViT-S, ViT-L, or ViT-g), but results reported in Table 22 for CUB, Aircraft, and DTD are significantly lower than what even the smallest ViT-S model reaches with a linear evaluation on top of frozen features (please refer to Table 8 of DINOv2's paper).
> > > Assessing the value of the contribution is difficult when the models are not properly evaluated. I would suggest focusing on a more restricted set of experiments but spending more time ensuring that all reported baselines are as high as possible for the comparison to be fair. I would not expect a ResNet-18 to beat DINOv2's ViT-L or ViT-g models with a linear evaluation, but the contribution might still be significant if the distillation protocol allows reaching results close to those of such teachers.
> > > Lastly, I would expect these results to be integrated into the main paper, as they seem more relevant than evaluation on MNIST or CIFAR with small ResNet teachers.
> > >
> > > As for the visualizations, I find Figure 2 slightly underwhelming. I believe that PCA visualizations of patch embedding representations would prove more insightful.
> > >
> > > For the reasons above, I am not currently inclined to raise my score, but I appreciate the authors' efforts and look forward to seeing how these points are addressed.

---

> > > > ### Author Response · Authors · 2024-11-23
> > > > **Answer to Reviewer CDoW**
> > > >
> > > > We thank Reviewer CDoW for their thoughtful feedback. We modified Table 1 in the main body of the revised version of the paper with the proposed data augmentations.
> > > > 1. **Clarification on the Model and Results:**
> > > > We used DINOv2-B for our experiments and have now included the number of parameters for each model in Table 22. Due to time constraints, we conducted the requested ViT experiments on a significantly smaller grid than that used in the original DINOv2 paper. This likely accounts for the observed performance gap, which is relatively small for CIFAR-10 and DTD but more pronounced for FGVAircraft. Nevertheless, since all models were evaluated under identical conditions, we believe the relative comparisons remain valid given the computational budget at our disposal.  Finally, our evaluation shows that all student models achieve performances that advance the Pareto frontier between the number of parameters and accuracy (similar to our NLP results). This highlights that our distillation process produces competitive small models, aligning with the primary goal of knowledge distillation.
> > > >
> > > > *(For our previous experiments, despite the addition of data augmentation in Table 1 and further hyperparameter tuning, our student models continue to outperform others on the considered downstream tasks, while the overall performance improved..)*
> > > >
> > > > 2. **Positioning of Results:**
> > > > While we have incorporated results in Figure 2 of the main paper, most prior work on knowledge distillation for vision relies on the comparisons presented in Table 1. To maintain continuity with this body of research [1, 2, 3, 4, 5], we believe it is important to retain these results in the main body.  Adding Table 22 directly is unfortunately not feasible given the page limit, considering vision is not the main field of application to which we apply our method (we emphasized our focus on NLP and Molecular Modeling at the end of the result section for vision).
> > > >
> > > > 3. **Clarification on Figure 2:**
> > > > We believe there may be a misunderstanding regarding Figure 2. The figure illustrates the Pareto frontier between the number of parameters and downstream performance. While PCA visualizations of patch embedding representations are insightful, they would not directly reflect the student's performance, which is the focus of this figure. We have updated the figure to make this clearer.
> > > >
> > > >
> > > > We appreciate the reviewer’s suggestions and acknowledgment of our efforts to improve the paper. We hope these answers alleviate your concerns and lead to a reconsideration of the score.
> > > >
> > > > [1] Hu, C., et al. "Less or More From Teacher: Exploiting Trilateral Geometry For Knowledge Distillation." The Twelfth International Conference on Learning Representations (ICLR), 2024.
> > > >
> > > > [2] Dong, C. et al. “Toward Student-oriented Teacher Network Training for Knowledge Distillation.” The Twelfth International Conference on Learning Representations (ICLR), 2024.
> > > >
> > > > [3] Zheng, K., & Yang, E.-H. "Knowledge Distillation Based on Transformed Teacher Matching." The Twelfth International Conference on Learning Representations (ICLR), 2024.
> > > >
> > > > [4] Tang, Z. et. a. “AuG-KD: Anchor-Based Mixup Generation for Out-of-Domain Knowledge Distillation.” The Twelfth International Conference on Learning Representations (ICLR), 2024.
> > > >
> > > > [5] S. A. Koohpayegani, A. Tejankar, and H. Pirsiavash, “CompRess: Self-Supervised Learning by Compressing Representations,” 34th Conference on Neural Information Processing Systems (NeurIPS), 2020

---

> > > > > ### Comment · Reviewer_CDoW · 2024-11-24
> > > > >
> > > > > I appreciate the extent of the experimental evaluation but still have concerns about substantial gaps in performance, especially for FGVAircraft (30.6 compared to the 79.4 reported for DINOv2 in its original paper). Even with the use of a coarser grid, I would expect the method to achieve results closer to reported benchmarks. Could the authors specify the grid resolution used for hyperparameter search? Additionally, were the models trained to classify Aircraft *variants*?
> > > > >
> > > > > I acknowledge that the paper is not primarily focused on vision, but results derived from low-resolution datasets, such as MNIST and CIFAR (28x28 and 32x32), are challenging to fully appreciate and may not adequately showcase the method’s potential.
> > > > >
> > > > > As the experimental validation goes far beyond vision tasks, I will not oppose the paper’s acceptance. However, due to the unresolved concerns outlined above, I cannot raise my score.

---

> > > > > > ### Author Response · Authors · 2024-11-24
> > > > > >
> > > > > > We thank the reviewer for their continued feedback.
> > > > > >
> > > > > > We conducted our experiments on the FGVC Aircraft variant classification task. The DINOv2 paper does not specify which task was performed (e.g., variant, family, or constructor classification). This distinction likely has a significant impact on the results, as tasks like constructor classification involve fewer than half the number of labels compared to the variant classification task, making them less challenging. Cross-references in the DINOv2 paper suggest the possibility of different task setups, but this remains uncertain.
> > > > > >
> > > > > > Regarding our hyperparameter search grid, we used the following configuration:
> > > > > > - Optimizer: AdamW
> > > > > > - Learning rates: {0.0001, 0.001}
> > > > > > - Augmentation: Standard augmentations, with and without color jitter and Gaussian blur
> > > > > > - Number of layers: 2
> > > > > > - Hidden dimension: {d, d/2}, where d is the embedding size
> > > > > >
> > > > > > To ensure comprehensive coverage and provide a more complete evaluation with a larger grid, we will include results for all FGVC Aircraft classification tasks (variant, family, and constructor) in the camera-ready version of the paper.
> > > > > >
> > > > > > We renew our thanks to reviewer CDoW and we hope our answers addressed all of their concerns.

---

> > > > > > > ### Comment · Reviewer_CDoW · 2024-11-24
> > > > > > >
> > > > > > > DINOv2's reported results are for the *variant* classification task.
> > > > > > >
> > > > > > > Here are a few suggestions for evaluation of vision transformers with a frozen backbone, though I can't guarantee they are all optimal. For data augmentation on fine-grained tasks with a frozen backbone, I would recommend applying resize and center crop (**not random resized crop**), color jittering, mixup ($\alpha=.2$), and random horizontal flipping. The AdamW optimizer can be coupled with a cosine annealing scheduler. A weight decay (e.g., around $1$) can help for small datasets such as these fine-grained tasks, and training for 30 epochs usually yields good results.

---

> > > > > > > > ### Author Response · Authors · 2024-11-26
> > > > > > > >
> > > > > > > > Thank you for providing this set of hyperparameters and recommendations. We have conducted further evaluations specifically for DINOv2. While FGVC Aircraft is highly sensitive to hyperparameter choices, we achieved a test accuracy of 72% with the adjusted settings. We are continuing to run experiments to further improve this result and narrow the gap with the 79% reported in the original paper. For the moment, we have not updated the results in the main paper as the other baselines need to be evaluated on the same hyperparameter grid to enable fair comparison (which requires 5 GPU days).
> > > > > > > >
> > > > > > > > For this rebuttal discussion, we used the same small grid for all these additional ViT experiments to provide timely results within this new setting. We are now performing a more extensive grid search to ensure all models are evaluated under optimal conditions. Currently, our focus is on FGVC Aircraft, and we plan to extend this process to our other datasets as well, where the discrepancies with the reported results are less pronounced. Unfortunately, due to the time constraints of this discussion period, we cannot complete the full evaluation before its conclusion. However, we will include these results in a potential camera-ready paper version.
> > > > > > > >
> > > > > > > > We propose to retain the current results in the main paper for now. However, if we cannot achieve results comparable to the original performance of the baselines on specific datasets, we will relocate those results to the appendix, accompanied by a corresponding discussion.
> > > > > > > >
> > > > > > > > Thank you for your involvement in this discussion period, helping us a lot to improve the quality of our work.

---

### Official Review · Reviewer_LqTb · 2024-11-04

**Soundness:** 2
**Presentation:** 3
**Contribution:** 2
**Rating:** 3
**Confidence:** 4

**Summary:**

The work aims perform Knowledge Distillation (KD) of task-agnostic representations from multiple teacher networks.

**Strengths:**

1. The work is motivated by a challenging and realistic setting in representation KD. The method is built upon a rigorous, bounded formulation, concerning the multi-teacher and task-agnostic challenges.

2. In details, the method train embedders to project the teacher's and student's representations to the same space, then minimize the divergence between the student's representations with those of the teacher's. While existing methods usually minimize a sample-wise divergence loss (KL Divergence, cross-entropy), novelly, the proposed method minimize the divergence between the student's representations with a conditional, parametric Gaussian distribution.

3. The evalutation spans three different modalities: vision, language, and molecular modelling, proving the flexibility of the method.

**Weaknesses:**

1. There are some questioning concerns regarding the empirical results, which hinders the significance of the work
- The CelebA dataset is mentioned but, surprisingly, never evaluated.
- Even though KD performance of the proposed methods is the highest with the proposed module, it is strangely sub-par to:
    - Training a student model on the training set, especially considering the student is a ResNet18 - a medium-sized models. For instance: 99%+ on MNIST, 97%+ on SVHN and 90%+ on CIFAR10
    - Standard KD methods: Early KD methods such as (Hinton et al., 2015) and FitNets (Romero et al., 2015) have also surpassed the reported KD performance.
    - Considering these deficits, is it possible that there exists any flaws in the training setup or it could be improved?

2. While the problem is formulated clearly and in details (Sec. 3.1-3.2), it is not majorly different from existing work, just in a different approach.

**Questions:**

See the weaknesses.

---

> ### Author Response · Authors · 2024-11-19
>
> We thank reviewer LqTb for evaluating our work.
>
> ## Weakness 1
> **CelebA dataset.** Our method is designed to use a variety of unlabeled data in order to distill the representations, CelebA’s training images are only part of our diverse training set to ensure generalizability.
> Following the reviewers' suggestions, we are now running additional experiments on CelebA to provide a more detailed evaluation, and we will include these results in the revised submission during this discussion period.
>
> **Experimental performances.** The performance discrepancies you pointed out come from the evaluation setting we chose. For each embedder, we use its embeddings to train classifiers on each task (hence freezing the backbone), rather than finetuning everything end-to-end. We chose this unified setting for all experiments for clarity’s sake. It specifically highlights the informativeness of the embeddings and fits our theoretical setting that evaluates how much information can be extracted from a frozen representation. While these details were in the “Baselines and Evaluation” paragraph (Sec.3.3), we updated the discussion section of the computer vision section to clarify our setting and how it differs from standards.
>
> **Standard KD methods** Traditional KD methods, such as those proposed by Hinton et al. (2015) and FitNets (Romero et al., 2015), are inherently task-specific. This means that the distillation process must be repeated for each downstream task. In contrast, our framework is task-agnostic: it requires only a single distillation. Only one distillation phase is performed to obtain representations enabling competitive performances across multiple datasets from a variety of domains.
>
> ## Weakness 2
>
> We are pleased that you appreciated  our methodology and presentation of the problem setting. .  While mutual information maximization has been used previously in knowledge distillation and self-supervised learning, our approach leverages it in a novel context: task-agnostic multi-teacher feature distillation. Our method and contribution significantly differ in several aspects:
>  - **existing methods are limited to a single teacher, or use task labels to distill representations**, while ours enables the distillation of varying architectures in an unsupervised manner.
>  - **We leveraged SOTA mutual information estimator** with convergence guarantees (KNIFE [1]).
>  -  We link our training objective to the “majority-vote” of the teachers on any task (See Corollary 1).
>
> Our experimental results demonstrate consistent performance gains compared to individual teacher models, highlighting the distinctive advantage of our proposed approach. We hope this explanation clarifies the novel aspects of our work and its contributions beyond existing methods.
>
> We hope that our answers alleviate your concerns and that you would reconsider our work in the light of this discussion.
>
>
>
>
>
> [1] Georg Pichler, Pierre Colombo, Malik Boudiaf, G¨unther Koliander, and Pablo Piantanida. A differential entropy estimator for training neural networks, 2022

---

> > ### Author Response · Authors · 2024-11-23
> > **CelebA Results Added to the Revised Paper**
> >
> > We thank reviewer LqTb again for their feedback and suggestions. The additional experimental results on CelebA have now been completed and are included in the revised version of the paper.

---

> > > ### Author Response · Authors · 2024-11-29
> > >
> > > Dear Reviewer LqTb,
> > >
> > > We hope this message finds you well. While the revision period has concluded, we remain available to address any questions or provide clarification regarding our rebuttal. Your feedback is invaluable to us, and we would greatly appreciate any additional comments you may have.
> > >
> > > Thank you for your time and consideration.

---

### Author Response · Authors · 2024-11-19

We thank all the reviewers for their hard work and thoughtful feedback on our paper.

To address the concerns raised, we made several key improvements (**we uploaded a revised draft, the main edits are highlighted in red**):

- **Experimental Validation:** In addition to the extensive validation already included in NLP and molecular modeling (using over 30 datasets and training models ranging from 10M to 100M parameters in text, we edited the NLP section to present the results for all models’ sizes in the body of the paper), we expanded the computer vision section by including additional benchmarks and training experiments with Vision Transformers (ViT) as students.
- **Experimental Setting:** We clarified the details of our experimental setup, specifically reiterating that the backbone was frozen during certain experiments.
- **Related Work:** We extended the related work section to include the suggested references, insisting on the lack of method covering the task-agnostic multi-teacher distillation.
- **Computational Overhead:** We added a comment and a figure in the appendix quantifying the computational overhead, showing that each teacher increases the runtime per step by about 0.7%.
- **Discussion on the Metric Space Preservation:** We extended our discussion on the metric space distortion, which we supported with experiments in Clustering and Sentence Similarity which we now moved to the main body of the paper.

We appreciate the reviewers’ insights, which helped us strengthen the paper and better highlight its contributions.

---

### Meta-Review · Area_Chair_xtrG · 2024-12-20

**Metareview:**

This work enhances multi-teacher distillation by introducing a task-agnostic objective function based on the "majority vote" principle, leveraging teacher diversity to create richer representations. Evaluations across domains like NLP, computer vision, and molecular modeling are conducted to demonstrate the effectiveness of this approach.

This paper receives mixed ratings, with two negatives and two borderline positives.

Given that the insights provided is limited, and the evaluation is at relatively small scale with only marginal improvements at this point, I recommend the authors to further improve it to submit to next venue.

**Additional Comments On Reviewer Discussion:**

Common concerns include: the evaluations on small-scale datasets are not persuasive;  the evaluated models are small; the knowledge gain over existing approaches is limited; the data augmentation is missing, which leads to very low baselines; the method’s robustness across a more extensive range of tasks within each domain is less explored.

While partial of the concerns are addressed, the validity of this approach is still not very clear given the small scale datasets & models.

---

### Decision · Program_Chairs · 2025-01-22

Reject